# Interaction of a viral insulin-like peptide with the IGF-1 receptor produces a natural antagonist

Francois Moreau [1,11], Nicholas S. Kirk [2,3,11], Fa Zhang[4], Vasily Gelfanov[5], Edward O. List[6], Martina Chrudinová[7], Hari Venugopal[8], Michael C. Lawrence [2,3], Veronica Jimenez[9,10], Fatima Bosch [9,10], John J. Kopchick [6], Richard D. DiMarchi [4], Emrah Altindis [7] & C. Ronald Kahn [1] ✉

Lymphocystis disease virus-1 (LCDV-1) and several other Iridoviridae encode viral insulin/IGF-1 like peptides (VILPs) with high homology to human insulin and IGFs. Here we show that while single-chain (sc) and double-chain (dc) LCDV1-VILPs have very low affinity for the insulin receptor, scLCDV1-VILP has high affinity for IGF1R where it can antagonize human IGF-1 signaling, without altering insulin signaling. Consequently, scLCDV1-VILP inhibits IGF-1 induced cell proliferation and growth hormone/IGF-1 induced growth of mice in vivo. Cryo-electron microscopy reveals that scLCDV1-VILP engages IGF1R in a unique manner, inducing changes in IGF1R conformation that led to separation, rather than juxtaposition, of the transmembrane segments and hence inactivation of the receptor. Thus, scLCDV1-VILP is a natural peptide with specific antagonist properties on IGF1R signaling and may provide a new tool to guide development of hormonal analogues to treat cancers or metabolic disorders sensitive to IGF-1 without affecting glucose metabolism.

The insulin/IGF superfamily of hormones includes insulin and insulin-like growth factors-1 and −2 (IGF-1 and 2), three isoforms of relaxin (RLN1, 2, and 3) and four insulin-like proteins (INSL3, 4, 5, and 6), which are expressed in development of specific tissues[1–8]. All of the members of this superfamily are synthesized as single chain precursors that share a similar primary structure consisting of a signal peptide followed by a prohormone with three domains: the A- and B-domains are localized at the C- and N-termini, respectively, with C-domain in between[9]. In addition, IGF-1, IGF-2 and INSL6 contain C-terminal extensions denoted as the D- and E-domains. In insulin, the C- domain is flanked by dibasic amino acid motifs (lysine and arginine) that are proteolytically processed, allowing secretion of the mature insulin double-chain disulfide-linked hormone and C-peptide[10,11]. IGF-1 and IGF-2, on the other hand, have short C-domains lacking these paired basic amino acids and thus are secreted as single chain hormones. All superfamily members also contain six highly conserved cysteine residues that form two interchain bonds between the A-domain and B-domain, as well as

[1]Section of Integrative Physiology and Metabolism, Joslin Diabetes Center, Harvard Medical School, Boston, MA, USA. [2]WEHI, Parkville, VIC, Australia. [3]Department of Medical Biology, Faculty of Medicine, Dentistry and Health Sciences, University of Melbourne, Parkville, VIC, Australia. [4]Department of Chemistry, Indiana University, Bloomington, IN, USA. [5]Novo Nordisk, Indianapolis Research Center, Indianapolis, USA. [6]Edison Biotechnology Institute and Heritage College of Osteopathic Medicine, Ohio University, Athens, OH, USA. [7]Boston College Biology Department, Chestnut Hill, MA, USA. [8]Ramaciotti Centre for Cryo-Electron Microscopy, Monash University, Clayton, VIC, Australia. [9]Department of Biochemistry and Molecular Biology, School of Veterinary Medicine and Center of Animal Biotechnology and Gene Therapy, Universitat Autonoma de Barcelona, Bellaterra, Spain. [10]CIBER de Diabetes y Enfermedades Metabólicas Asociadas (CIBERDEM), 28029 Madrid, Spain. [11]These authors contributed equally: Francois Moreau, Nicholas S. Kirk. ✉ e-mail: c.ronald.kahn@joslin.harvard.edu

one intrachain disulfide bond in the A-domain[11]. In addition, all the members, for which the tertiary structure was determined, exhibit similar features in which the A-chain forms a peptide with two α-helices and the B-chain forms a peptide containing one α-helix with

non-helical regions at both the N- and C- termini whereas the other members have a similar predicted structure[11] (Fig. 1a, c).

Insulin, IGF-1, and IGF-2 signal through two highly homologous receptor tyrosine kinases, the insulin and IGF-1 receptors (IR and IGF1R,

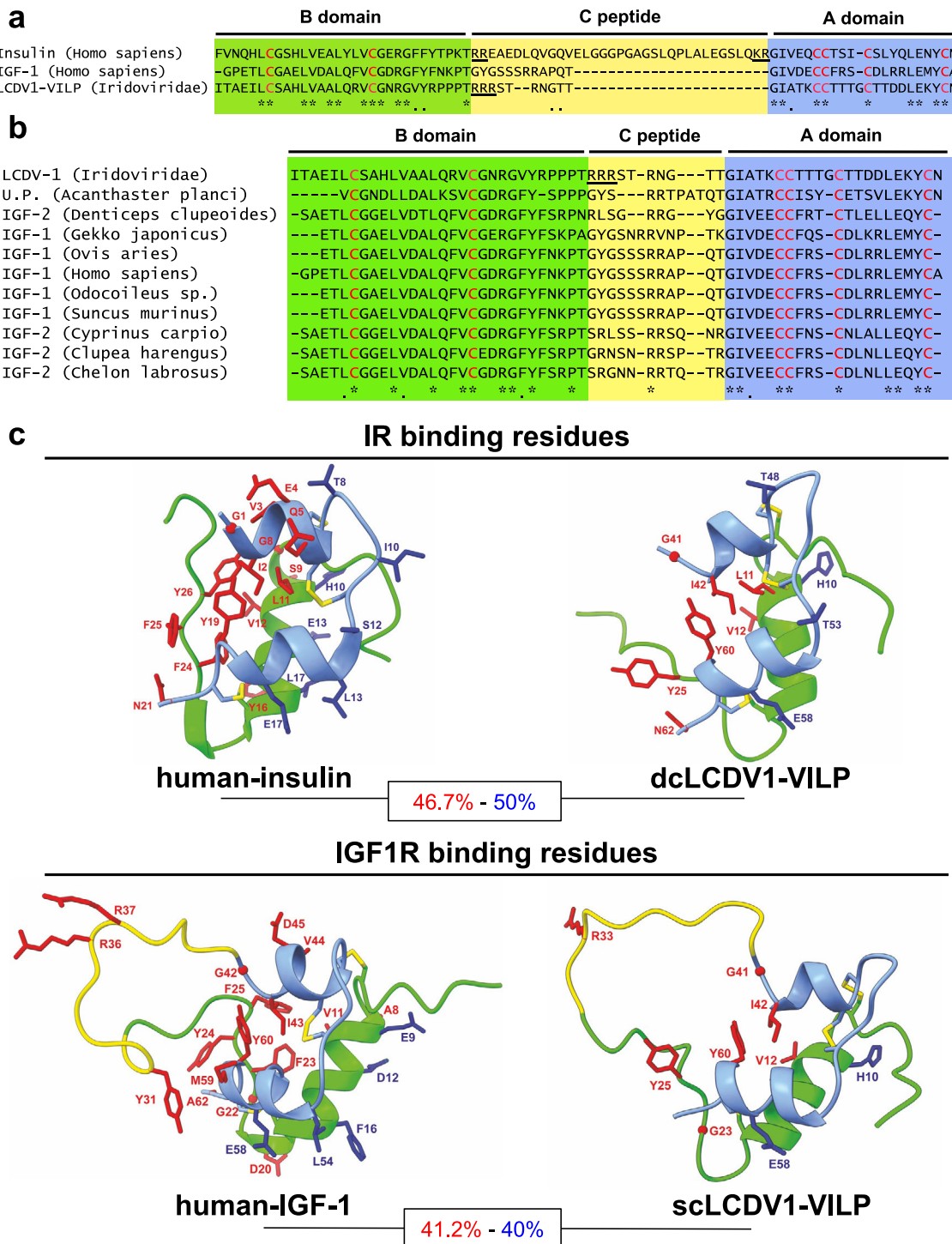

**Fig. 1 | Structural homology of LCDV1-VILP with insulin and IGFs. a** Sequence alignment of LCDV1-VILP with the human insulin and IGF-1. Strictly conserved cysteine residues are in red. The yellow square represents the C-domain, while the A- and the B-domains are in blue and green respectively. Cleavage sites composed of dibasic residues are underlined. Conserved amino acids are represented by an asterisk (*), and conservatively substituted residues are noted as a dot (.). **b** The LCDV1-VILP sequence was aligned with the 10 sequences showing the highest sequence identity. **c** Three-dimensional representation of insulin (PDB: 3I40), IGF-1

(AlphaFold2 prediction, pLDDT 78.23) and LCDV1-VILP as a single- (sc) and double chain (dc) molecules (AlphaFold2 predictions, pLDDT scores: scLCDV1 = 71.47, dcLCDV1 = 80.58). The A- and B-chains are represented in blue and green, respectively, and the C-peptide is in yellow. Residues involved in binding to site 1 and site 2 of the insulin/IGF-1 receptor are respectively in red and blue. The percentage of conserved binding residues with the sites 1 and 2 are indicated in red and blue, respectively.

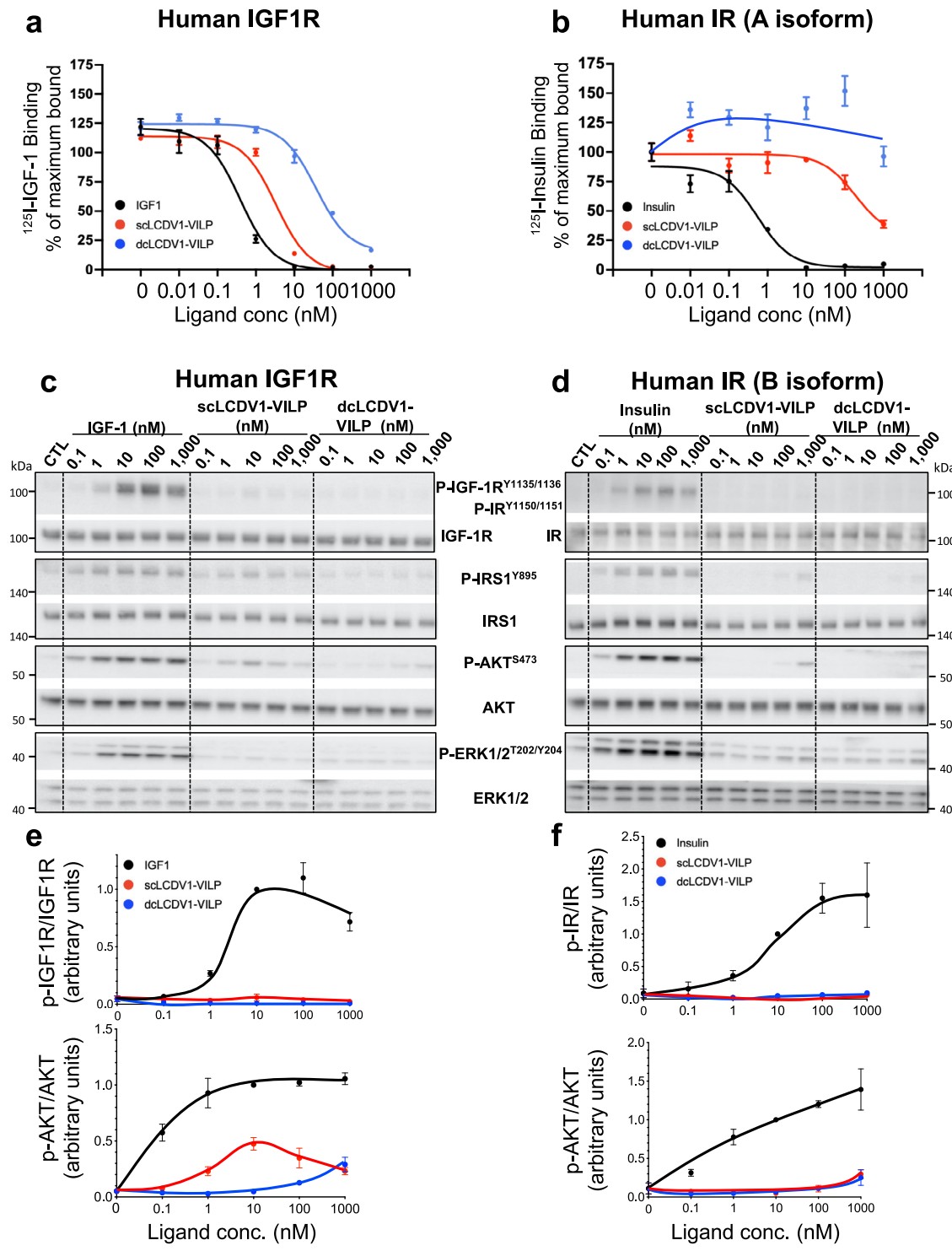

**Fig. 2 | LCDV1-VILP as a single chain peptide is more potent to bind on IGF1R and on IR.** Competition binding assays on human IGF1R (**a**) and human IR (**b**) ectodomains. Data are expressed as the mean ± SEM, normalized to the baseline and expressed as % of the maximal binding of $^{125}$I-IGF-1 (**a**) and $^{125}$I-Insulin (**b**) incubated alone (n = 3 (IGF1R) and 4 (IR) independent wells). Western blot detection of IGF1R, IR, AKT, IRS1, ERK1, and ERK2 phosphorylation in lysates of murine preadipocytes overexpressing the human IGF1R (**c**) or the human IR (**d**). Quantitative analysis of phosphorylated IGF1R, IR and AKT. Data, expressed as mean ± SEM (n = 3 independent experiments), were normalized by phosphorylation intensity induced by 10 nM of IGF-1 (**e**) or insulin (**f**).

respectively)[12–14], whereas the relaxins and INSL peptides act through G-protein coupled receptors[15]. The human IR exists in two isoforms (A and B), which differ by 12 amino acids at the C-terminus of the A-chain due to alternative splicing of exon 11 (A isoform: -exon 11; B isoform: +exon11), whereas human IGF1R exists as a single isoform[16]. Insulin binds with high affinity to both the A- and B-isoforms of IR but exhibits much lower affinity for IGF1R[17,18]. Conversely, IGF-1 shows a high affinity

for the IGF1R and a low affinity for the IR[19], whereas IGF-2 has relatively high affinity for both the IGF1R and the A-isoform of IR[17]. Structural studies have revealed that in their ligand-free (apo) form, the extracellular domain of these receptors form an inverted "V" (or inverted "U")[20] that holds their transmembrane and intracellular elements apart, about 67 Å in the case of IGF-1 receptor and 115 Å in the case of IR. The binding of IGF-1 to the IGF-1 receptor (or insulin to IR) then

leads to a conformational change that permits the receptor "legs" to come together, enabling transphosphorylation of the intracellular tyrosine kinase domains and activation of the receptor[21].

Insulin and IGFs have been found in animal species ranging from insects to humans[22]. Recently, we extended the scope of the insulin/IGF family of hormones by identifying viral insulin/IGF-1 like peptides (VILPs) in four members of the Iridoviridae family of double-stranded-DNA viruses[23]. While Iridoviridae primarily infect fish, reptiles and insects[24], DNA of these VILP-carrying viruses have been found in the human fecal virome[25,26] and in human blood[27], indicating potential exposure of humans to these VILPs. These VILPs show high sequence homology to insulin, IGF-1, and IGF-2[23]. They also possess short IGF-like C-peptide regions, but in contrast to human IGF-1/2, some VILPs have potential sites for cleavage, raising the possibility that they may act as single- and/or double chain molecules depending on the infected cells expressing them. In previous studies we have shown that when synthesized as single chain (sc) molecules, VILPs can bind to and activate both human IR (hIR) and human IGF1R (hIGF1R), with higher affinity for hIGF1R. Single-chain VILPs can also stimulate proliferation of human fibroblasts, increase glucose uptake into mouse adipocytes in vitro, and, at high dose, decrease blood glucose in vivo in mice[23]. In addition, VILPs can have unique properties not observed with insulin or IGF1. For example, when synthesized as double-chain (dc) peptides, the VILPs from Singapore Grouper Iridovirus (SGIV) and Grouper Iridovirus (GIV) not only act as insulin analogs in vitro and in vivo, but dcGIV-VILP also exhibits preferential tissue-specific action to stimulate glucose uptake in white adipose tissue[28]. Also recently, we have shown that Lympho-cystis disease virus 1 (LCDV1) VILP can inhibit hIGF1R phosphorylation in vitro[29] suggesting it may have antagonistic actions on this receptor.

In the present study, we have characterized the role of sc- and dcLCDV1-VILPs as ligands of the hIR and hIGF1R in vitro and in vivo. We find that both sc- and dcLCDV1-VILPs are very weak IR agonists. scLCDV1-VILP, on the other hand, is a weak partial agonist and potent competitive antagonist of IGF1R. More interestingly, scLCDV1-VILP can inhibit IGF-1-induced signaling, reduce IGF-1-induced cell proliferation in vitro, and slow body weight gain of transgenic mice with high cir-culating levels of IGF-1 due to over-expression of bovine growth hor-mone (bGH). Using cryo-electron microscopy (cryoEM), we find that scLCDV1-VILP in complex with the IGF1R ectodomain forms a unique interaction at the ligand-binding site and a dramatically different overall receptor conformation, explaining scLCDV1-VILP's antagonism. Thus, scLCDV1-VILP presents a new naturally occurring antagonist of the IGF1R and understanding this ligand-receptor interaction may also assist design of analogues that could serve as treatment for metabolic and growth disorders linked to hyperactivation of IGF1R signaling, with a minimal impact on glucose metabolism.

## Results
### LCDV1-VILP shares a similar structure with vertebrate and invertebrate IGFs and insulins
All members of the insulin/IGF superfamily, including VILPs, exhibit comparable domain architecture (Fig. 1a)[11,23]. Sequences of the ten insulin/IGFs showing the highest percentage of similarity to LCDV1-VILP are aligned in Fig. 1b and can be found in the Supplementary Data 1. Among these sequences, starfish insulin-like peptide, herring IGF-2 and mammal insulin/IGFs showed the highest sequence iden-tity (about 50%) with LCDV1-VILP. Like human IGF-1 and −2, LCDV1-VILP has a short C-peptide (10 amino acids); however, unlike these IGFs, this C-peptide contains only one site of potential cleavage represented by three successive arginines at the B-domain/C-domain junction (underlined in Fig. 1b), which might be recognized by enzymes such as the proprotein convertases 1 and 2[30]. Thus, depending on the cellular host and its potential for processing and secretion, LCDV1-VILP could be produced as a single chain or double chain peptide.

**Table 1 | Receptor binding IC$_{50}$ of human IGF-1, human insulin, and LCDV1-VILPs**

| Ligands | IC$_{50}$ (nM) | |
|---|---|---|
| | IGF1R | IR |
| IGF-1 | 0.4 | – |
| Insulin | – | 0.5 |
| scLCDV1-VILP | 3.3 | 188.7 |
| dcLCDV1-VILP | 36.3 | >1000 |

Single-chain (sc) and double-chain (dc) LCDV1-VILP are predicted to share similar tertiary structures to IGF-1 and insulin (Fig. 1c). Sequence alignment between human insulin and dcLCDV1-VILP revealed that 7 out of 15 residues involved in insulin's binding to site 1 in the insulin receptor (shown in red) and 4 out of 8 residues involved insulin's binding to site 2 (shown in blue) are conserved in LCDV1-VILP. Similarly, 7 out of 16 and 2 out of 5 of the residues involved in IGF-1's binding to IGF1R sites 1 and 2, respectively, are conserved in scLCDV1-VILP[23].

### scLCDV1-VILP is more potent in binding to IGF1R than IR
To understand the differences in the two potential forms of LCDV1-VILP, we synthesized both the single chain (more IGF-1-like) and the double chain (more insulin-like) peptides and assessed their ability to compete with $^{125}$I-insulin and $^{125}$I-IGF-1 for binding to the ectodomains of the hIR (A-isoform) and hIGF1R. As expected, unlabeled IGF-1 was able to compete with labeled IGF-1 for binding to IGF1R with an IC50 (i.e., the concentration of unlabeled ligand that inhibits labeled ligand binding by 50%) of 0.38 nM (Fig. 2a and Table 1). Compared to hIGF-1, scLCDV1-VILP was about an order of magnitude less potent in com-peting for binding to the IGF-1 receptor, but >10-fold more potent than dcLCDV1-VILP (IC50 = 3.3 nM vs. 36.6 nM). scLCDV1-VILP was also more potent in competing with $^{125}$I-Insulin for binding to IR than dcLCDV1-VILP with IC50s of ~188 nM and >1000 nM, respectively, although both VILPs had very low affinity compared to insulin itself, which had an IC50 of 0.55 nM (Fig. 2b and Table 1).

To assess the ability of sc- and dcLCDV1-VILPs to activate the IR and IGF1R signaling pathways, we used murine brown preadipocyte cell lines in which the endogenous IR and IGF1R genes had been inactivated to create a double knockout cell, after which the cells were reconstituted with either hIGF1R alone or hIR (B isoform) alone. In preadipocytes over-expressing hIGF1R, IGF-1 stimulated receptor autophosphorylation in a dose-dependent manner with half-maximal stimulation at 3.8 nM (Fig. 2c, e). On the other hand, despite its ability to compete for binding to IGF1R with about 10% of the affinity of IGF-1, scLCDV1-VILP only minimally induced autophosphorylation of IGF1R, and this peaked at 10 nM ligand and then decreased with further increases in ligand concentration. Consistent with its lower affinity for IGF1R, dcLCDV1-VILP produced no measurable increase in IGF1R autophosphorylation.

Similar dose response curves were seen for stimulation of IRS-1 Tyr$^{895}$ phosphorylation and AKT Ser$^{473}$ phosphorylation for both sc- and dcLCDV1-VILPs. Again, with the single-chain ligand, IRS-1 and AKT phosphorylation were much lower than with IGF-1, reaching a max-imum at 10 nM ligand then decreasing at higher concentrations, sug-gesting potential inhibition of signaling at these higher concentrations. Interestingly, in cells expressing hIGF1R, neither sc- nor dcLCDV1-VILP stimulated any detectable ERK1/2$^{T202/Y204}$ phosphorylation, indicating a unique post-receptor pathway bias in the signaling by these ligands. Since IGF1R undergoes internalization following its stimulation, we also compared the ability of scLCDV1-VILP and IGF1 to stimulate IGF1R endocytosis, using cells in which we had biotinated surface proteins[31]. As expected, stimulation of cells expressing hIGF1R with 10 nM IGF1 lead to a significant decrease by 57% of IGF1R at the membrane by

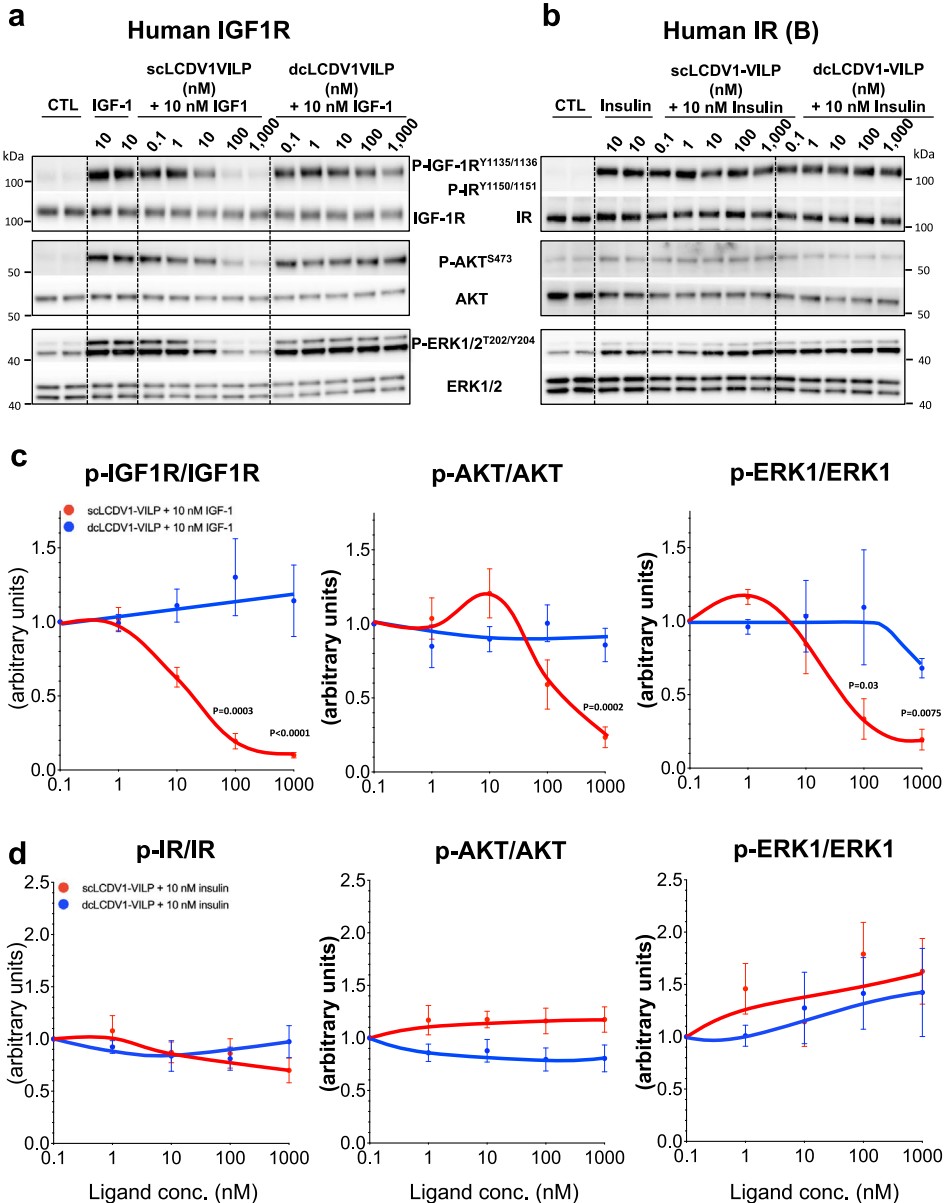

**Fig. 3 | scLCDV1-VILP is a potent IGF1R antagonist.** Western blot detection of IGF1R, IR, AKT, ERK1 and ERK2 phosphorylation in lysates of murine preadipocyte overexpressing the human IGF1R (**a**) or the human IR isoform B (**b**) as described in Methods. Quantitation of phosphorylated IGF1R, IR, AKT and ERK1. Scan intensities were normalized to total protein, and data expressed as mean ± SEM (two-way ANOVA followed by a Šídák's multiple comparison test; Graphpad Prism V.9; $n = 4$ (IGF1R) and 6 (IR) independent experiments; data were normalized by phosphorylation intensity induced by 10 nM of IGF-1 (**c**) or insulin (**d**).

45 min (Supplementary Fig. 1a, b). At the same time point, scLDV1-VILP at 10 and 1000 nM were able to reduce the level of membrane IGF1R by 40% and 47% respectively, in line with its lower potency to bind and activate IGF1R.

In double knockout preadipocytes expressing only the human IR, insulin produced an expected dose-dependent increase in receptor autophosphorylation, but neither sc- nor dc-LCDV1-VILP had a measurable effect. In these cells, single chain-, but not double chain-, LCDV1-VILP did produce a very low level of IRS-1$^{Y895}$ and AKT$^{S473}$ phosphorylation at the highest concentrations tested, and again neither had an effect on ERK1/2$^{T202/Y204}$ phosphorylation (Fig. 2d, f).

**scLCDV1-VILP is a potent IGF-1 receptor antagonist**
The above data indicate that while scLCDV1-VILP can bind to hIGF1R with relatively high affinity, it has very little signaling capacity, suggesting it may be a competitive antagonist. To examine this hypothesis in more detail, we performed a series of co-stimulation experiments using both sc- or dcLCDV1-VILP along with IGF-1 or insulin and murine preadipocytes overexpressing the hIR or hIGF1R. In cells expressing only the hIGF1R, 10 nM IGF-1 produced a robust (>40-fold) stimulation of IGF1R$^{Y1135/Y1136}$ autophosphorylation, and this was accompanied by increased downstream signaling with >10-fold increases in phosphorylation of AKT$^{S473}$ and ERK1/ERK2$^{T202/Y204}$ (Fig. 3a). Interestingly, when IGF-1 (10 nM) was co-incubated with an equal concentration of scLCDV1-VILP, autophosphorylation of IGF1R$^{Y1135/Y1136}$ and phosphorylation of AKT$^{S473}$ and ERK1/ERK2$^{T202/Y204}$ were reduced by ~40%. Further increasing the concentration of scLCDV1-VILP to 100 and 1000 nM increased this inhibition, with almost a complete blockade IGF-1 stimulated receptor autophosphorylation, as well as phosphorylation of AKT$^{S473}$ and ERK1/ERK2$^{T202/Y204}$ at the highest VILP concentration (Fig. 3a, c). By contrast, co-incubation with dcLCDV1-VILP had no effect on IGF-1 stimulation of receptor autophosphorylation or downstream signaling. This antagonistic effect of scLCDV1-VILP was specific to IGF1R. Thus, using DKO cells expressing the human IR, addition of

10 nM insulin marked increased IR$^{Y1150/Y1151}$ autophosphorylation and stimulated AKT$^{S473}$ and ERK1/ERK2$^{T202/Y204}$ phosphorylation (Fig. 3b). In contrast to the effects on IGF1R signaling, however, co-incubation of these cells with either sc- or dcLCDV1-VILP at concentrations from 0.1 to 1000 nM had no significant effect on phosphorylation on insulin stimulation of IR$^{Y1150/Y1151}$ or its downstream kinases (Fig. 3b, d). Insulin/IGF-1 signaling in response to either IGF-1, sc- or dcLCDV1-VILP, was assessed in MCF-7 cells, expressing endogenous levels of both hIR and hIGF1R. Consistent with our data shown in (Fig. 2a, c) IGF-1 was more potent than sc- and dcLCDV1-VILP in inducing phosphorylation of the IGF1R, AKT and ERK1/ERK2 (Supplementary Fig. 2a). In line with the competition binding assay performed on hIGF1R (Fig. 2a), scLCDV1-VILP was more potent than dcLCDV1-VILP in stimulating AKT phosphorylation. Consistent with the Fig. 3a, c, only the scLCDV1-VILP was able to antagonize IGF1R signaling induced by 10 nM of IGF1.

## scLCDV1-VILP inhibits IGF-1 induced cell proliferation and mouse growth

A major effect of IGF-1 is stimulation of cell cycle progression and cell replication[32]. In murine brown preadipocyte over-expressing hIGF1R, hIGF-1 produced a dose-dependent increase in cell number with a half-maximal effect at ‐0.07 nM IGF-1 and a maximal effect at 1 nM (Fig. 4a). At low concentrations, scLCDV-VILP also induced cell proliferation, but the curve was shifted one-order of magnitude to the right, peaking at 10 nM and then decreasing to basal levels with further increases in ligand, suggesting a self-inhibitory effect. This antagonistic effect of scLCDV1-VILP was even clearer when preadipocytes expressing hIGF1R were incubated with 10 nM of IGF-1 and increasing concentrations of scLCDV1-VILP. Thus, consistent with the signaling data, co-incubation of IGF-1 with increasing concentrations of scLCDV1-VILP produced a dose dependent inhibition of IGF-1 stimulated cell proliferation (Fig. 4b). By contrast, dcLCDV1-VILP had no effect on IGF-1 mediated cell proliferation (Fig. 4a, b). The lack of increase in cell number was not related to increased apoptosis, since cell survival was not changed by high dose of either scLCDV1-VILP or dcLCDV1-VILP (Fig. 4c).

To assess the functional consequence of LCDV1-VILP antagonism of IGF1R signaling in vivo, we utilized Adeno-Associated Vector serotype 8 (AAV8) viral vectors carrying the sequence of LCDV1-VILP and a transgenic mouse model expressing bGH. These mice have been previously shown to have elevated circulating levels of IGF-1, an increase in lean mass, accelerated growth, and reduced lifespan compared to wildtype littermates[33,34]. As expected, transfection of mice with AAV-LCDV1-VILP led to a dramatic increase in the levels of *Lcdv1-vilp* mRNA in liver compared to undetectable levels in mice receiving the control AAV (Fig. 4d). This was associated with a slight, but not significant, increase in the expression of endogenous *Igf-1* in liver, as well (Fig. 4d). Nonetheless, over the following seven weeks, body weight gain in the bGH transgenic mice that received the AAV-LCDV1-VILP was significantly reduced versus those receiving control AAV (Fig. 4e, f). Nuclear magnetic resonance revealed that this reduction of weight gain was accompanied by a reduction in lean body mass, but no change in fat mass (Supplementary Fig. 3a–c). There was also no change in the levels of fasting blood glucose (Fig. 4g).

## CryoEM modeling of the IGF1R ectodomain in complex with scLCDV1-VILP

To determine how LCDV1-VILP effects antagonism of hIGF1R, we performed cryo-electron microscopy (cryoEM) imaging of an IGF1R ectodomain construct IGF1Rzip[35] prepared in the presence of a 30-fold stoichiometric ratio of scLCDV1-VILP (~30-fold molar excess calculated per receptor αβ monomer). The initial reconstruction was readily interpretable, displaying a pseudo-two-fold symmetric arrangement of IGF1Rzip domains, with each L1-CR + αCT′ module engaging a scLCDV1-VILP moiety. Further 'non-uniform' refinement (NU-refinement) in cryoPARC (v3.2) led to a final map with improved resolution

(GS-FSC 0.143 *ca* 4.6 Å), sufficient to permit unambiguous placement of the scLCDV1-VILP moieties and the domains of the receptor. Whereas the resolution varied significantly across the map (local resolution: min 2.4 Å, median 6.8 Å, 75th percentile 8.4 Å), highest resolution was achieved at one scLCDV-1 VILP binding site, permitting conservative morphing of an AlphaFold2[36] (AF2) model of scLCDV1-VILP to fit the density. The resultant model and associated cryoEM potential density are shown in (Fig. 5a–c, Supplementary Fig. 4e).

Within our atomic model, the structural changes within each receptor monomer upon scLCDV1-VILP binding (Supplementary Fig. 4a) are seen to be considerably less marked than those induced by IGF-1 binding to the IGF-1R ectodomain (Supplementary Fig. 4b). The L1 domain is displaced from its apo location adjacent to domain FnIII-2′ by only ~13 Å, compared to ~42 Å in the IGF-1 bound case (Fig. 5d, e). Also, although difficult to quantify due to heterogeneity in the relative disposition of domains FnIII-3 and FnIII-3′, the points of membrane entry are judged here to be approximately 90 Å apart, based on a linear extrapolation from domains FnIII-1 and FnIII-1′. Such separation is greater than that in the apo IGF1R ectodomain and likely incompatible with signaling activation[21].

In contrast to what is seen within the IGF-1 bound receptor[37], the scLCDV1-VILP moieties are not translocated to the receptor head module and do not engage the membrane-distal regions of the FnIII-1 or FnIII-1′ domains (Fig. 5f, g vs Fig. 5h). Instead, the scLCDV1-VILP bound to domain L1 makes a sparse contact with domain FnIII-1′ in the vicinity of residue His539′. This residue lies towards the C-terminus of canonical strand E of domain FnIII-1′ and, based on the homology of IGF1R and IR, may form part of IGF-1's binding site 2 (equivalent to IR Arg554′)[38]. The [domain-L1′]-bound scLCDV1-VILP moiety approaches, but however does not engage, domain FnIII-1 (Fig. 5f, g).

The C peptide of scLCDV1-VILP appears more ordered in our structure than that of IGF-1 in its receptor complex[37] and is displaced more towards the C-terminal end of the domain-L1 β barrel (Supplementary Fig. 4c vs Supplementary Fig. 4d). We note the substitution of Gly7 in IGF-1's B-chain (a highly conserved residue in the insulin/IGF-1 superfamily) by serine in LCDV1-VILP (Fig. 1a) as the presence of serine (rather than glycine) at this location is important for the antagonistic properties of LCDV1-VILP[29]. Glu9 in the B-chain of IGF-1 is a key residue involved in the binding with IGF1R[39]. In LCDV1-VILP, Glu9 is replaced by histidine, which would alter this interaction.

## Discussion

Viral insulin-like peptides (VILPs) represent a unique branch of the insulin/IGF-1 superfamily tree, being the only insulin-like molecules in non-eukaryotic organisms. In this study we have explored the actions of one of these unique insulin-like molecules, LCDV1-VILP, as a single-chain (sc) IGF-1-like and double-chain (dc) insulin-like peptide, as well as interaction of scLCDV1-VILP with IGF1R receptor at a structural level. We find that despite its insulin-like structure, dcLCDV1-VILP has no ability to interact with the human insulin receptor and no insulin-like actions. The IGF-1-like single-chain LCDV1-VILP, on the other hand, has relatively high affinity for hIGF1R (‐12% that of hIGF-1). However, in contrast to IGF-1, scLCDV1-VILP has very low agonist activity (<1%), but instead acts as a potent competitive antagonist of IGF1R both in vitro and in vivo. This action of scLCDV1-VILP is due to a different mode of interaction with IGF1R. Thus, in contrast to IGF-1 that binds in a 1:1 ligand to receptor ratio, two molecules of LCDV1-VILP bind to the receptor dimer, but they are unable to induce the conformational change to allow site 1b interaction, and as a result, the extracellular juxtamembrane domains, and hence the associated transmembrane and intracellular domains, move further apart rather than closer together, leading to an inability of the IGF1R kinase to undergo transphosphorylation and activation of the receptor toward its downstream substrates.

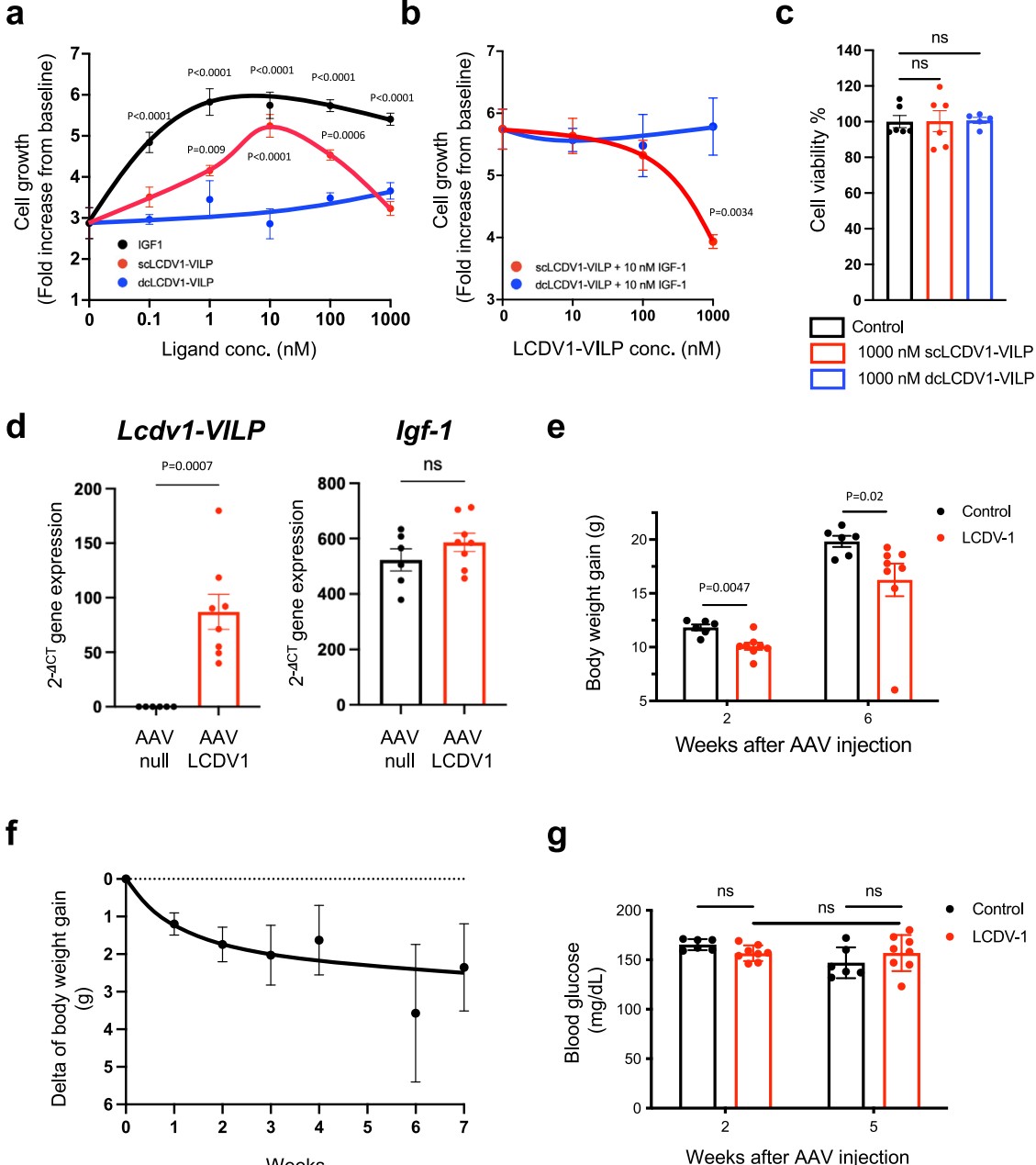

**Fig. 4 | scLCDV1-VILP reduces cell proliferation and growth in mice over-secreting bovine growth hormone. a** Murine preadipocytes overexpressing the human IGF1R were plated at a density of 20,000 cells/well then counted after a 48-h incubation with the indicated concentrations of IGF-1, scLCDV1-VILP and dcLCDV1-VILP. Data are expressed as the mean ± SEM (Two-way ANOVA followed by a Dunnett's multiple comparison test (Graphpad Prism V.9); $n$ = 3 independent experiments). **b** The number of murine preadipocytes expressing the hIGF1R were counted after a 48-h incubation with various concentrations of scLCDV1-VILP and dcLCDV1-VILP and a fixed concentration of IGF-1 (10 nM). Data are expressed as the mean ± SEM (two-way ANOVA followed by a Šídák's multiple comparison test (Graphpad Prism V.9); $n$ = 3 independent experiments). **c** Viable murine preadipocytes overexpressing the hIGF1R were stained with crystal violet after a 48-h incubation with various concentrations of IGF-1, scLCDV1-VILP and dcLCDV1-VILP. Data are express as mean ± SEM (ns, not significant; Kruskal-Wallis test followed by

a Dunn's multiple comparisons test (Graphpad Prism V.9); $n$ = 5 (scLCDV1-VILP) and 6 (dcLCDV1-VILP) independent wells). **d** Relative expression of the *LCDV-1* and *IGF-1* mRNA expression in liver samples from mice that received either an AAV control ($n$ = 6 mice) or the AAV LCDV-1 ($n$ = 8 mice). Data are expressed as the mean ± SEM (ns, not significant; two-sided Mann–Whitney test; Graphpad Prism V.9). **e** Body weight gain at week 2 and 6 following AAV injection. Data are normalized by the basal weight measured at $t$ = 0 and expressed as mean ± SEM (two-sided Mann–Whitney test; Graphpad Prism V.9; AAV control = 6 mice, AAV LCDV-1 = 8 mice). **f** The increment (delta) of body weight gain was measured weekly for seven weeks following the AAVs injection. Data are express as mean ± SEM (AAV control=6 mice, AAV LCDV-1 = 8 mice). **g** Fasting blood glucose was assessed 2 and 5 weeks after the AAVs injection. Data are express as mean ± SEM (ns, not significant; two-way ANOVA followed by a Šídák's multiple comparison test; Graphpad Prism V.9; AAV control=6 mice, AAV LCDV-1 = 8 mice).

Interestingly, while viruses are very diverse and include both RNA and DNA carrying species in over 180 families, all of the VILPs discovered thus far are encoded by double-stranded DNA viruses belonging to the Iridoviridae family[23]. This family of viruses is known to infect primarily fish, reptiles and insects[24]. Although, LCDV1-VILP

shares ~50% homology with the IGFs of certain species of fish and reptiles, which might suggest horizontal gene transfer, the sequence conservation of LCDV1-VILP is almost as high with human IGFs and insulin (~47%), making this hypothesis uncertain. The exact role of these VILPs in viral function remains to be determined, infected fish

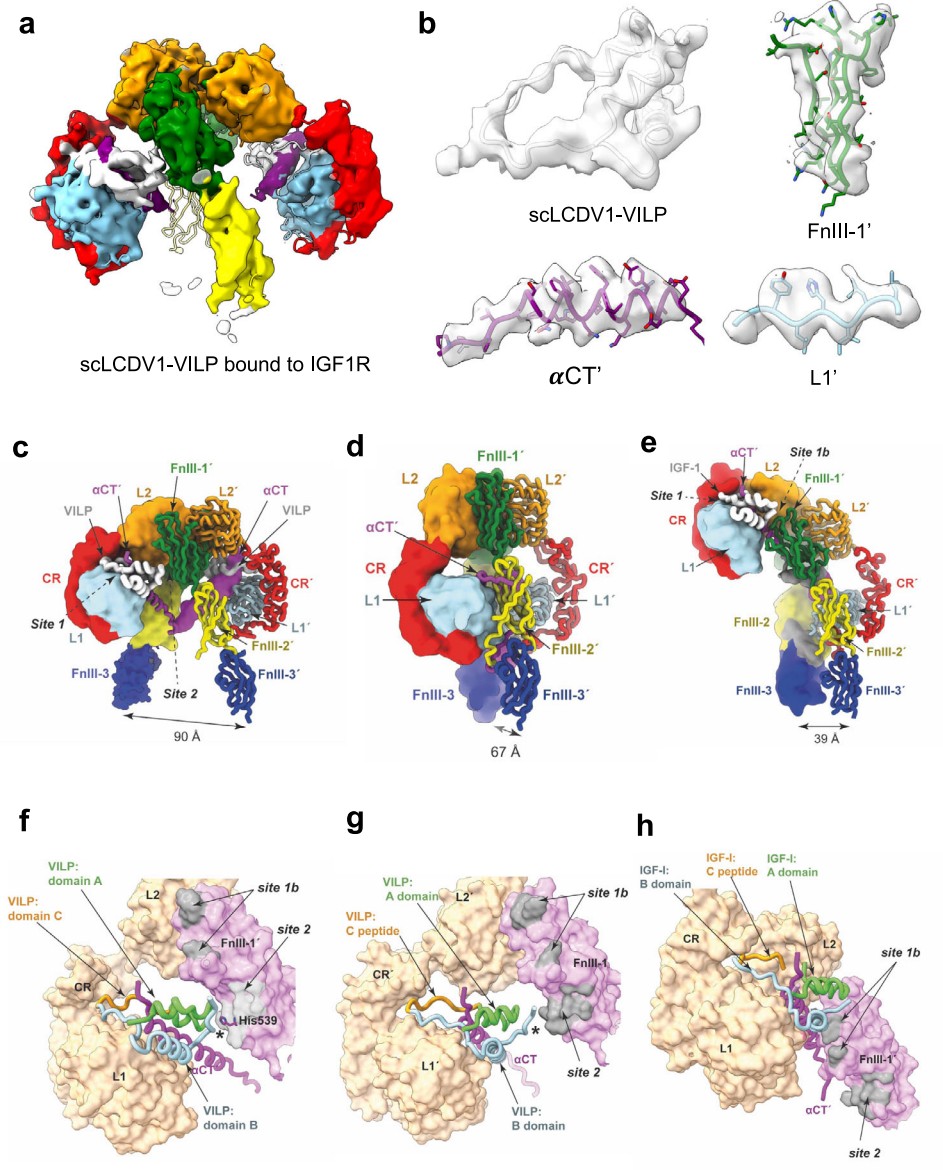

**Fig. 5 | Relative conformations of the apo- and ligand-bound IGF1R ectodomain. a** CryoEM density plus the atomic model of scLCDV-1VILP bound to IGF1R.zip shown in ribbon representation (map contour level 0.149). **b** CryoEM density associated with scLCDV1-VILP, domain FnIII-1′, αCT′ and domain L1 (map contour level 0.149). **c** CryoEM structure of IGF1R ectodomain in complex with two copies of scLCDV1-VILP with one receptor monomer in low-resolution surface representation and the other in ribbon representation. Note domains FnIII-3 and FnIII-3′ and upstream regions of the insert domains are positioned based on weak potential density from an initial consensus cryoEM map; these entities are unmodelled in the final deposited structure as they are not discerned in the final potential map.

Domains are labeled L1, L2: first and second leucine-rich repeat domains; FnIII-1, -2, -3: first, second and third fibronectin Type III domains; αCT: C-terminal α-helical segment of IGF1R α-chain (prime symbol ′ denotes domains from alternate monomer). **d** Apo IGF-1R ectodomain crystal structure[20] (PDB: 5U8R). Domain representation and nomenclature is as in panel **c. e** CryoEM structure of a single IGF-1 bound to IGF1R (PDB: 6PYH)[37]. Domain representation and nomenclature as in **c. f, g** Relative disposition of scLCDV1-VILPs and (**h**) IGF-1 upon binding to receptor, illustrating (*) the comparative lack of interaction of the L1′-bound scLCDV1-VILP with domain FnIII-1.

develop wart-like nodules on their fins, skin, or gills, which represent clusters of fibroblasts massively enlarged due to viral production. Although not believed to be a human pathogen, it is important to note that sequences of the LCDV-1 virus have been found in the human fecal virome[25,26] and in human blood[26], suggesting that humans are likely exposed to these viral insulin-like peptides. Furthermore, scLCDV1-VILP has a high affinity for both the human and murine IGF-1 receptors[23], indicating that if humans are exposed to this viral peptide, it may have biological consequences[40].

Whether LCDV1-VILP or other VILPs are processed to double chain, insulin-like molecules or remain as single chain, IGF-1-like

molecules, in cells infected with these viruses remains unknown. Two of the major differences between vertebrate insulins and IGFs are in the length of C-peptide region that links the B- and A-domains (30-33 amino acids in insulins versus 8-12 amino acids in IGFs) and the presence of dibasic amino acids at the C-peptide junctions with the A- and B-chains in proinsulin, allowing the C-peptide to be cleaved out, leaving the mature two-chain insulin molecule[11]. LCDV1-VILP is a hybrid of these features, having a short IGF-like C-peptide (10 amino acids), but having basic amino acids at the theoretical B-chain/C-peptide junction, which potentially allows for proteolytic processing to a two-chain molecule in LCDV1-infected cells. To understand the differences

between these two products, in the present study we have compared the actions of scLCDV1-VILP with that of dcLCDV1-VILPs both in vitro and in vivo.

While the differences between sc- and dcLCDV1-VILPs are modest on insulin receptor signaling, since they have very low affinity for the IR, they are dramatic on the IGF-1 receptor. Thus, scLCDV1-VILP has an affinity for IGF1R that is almost 12% that of hIGF-1, as compared to the double chain hormone, which has <1% of the affinity of the native ligand. This is likely due to its more IGF-1-like structure. In addition, the C-domains of LCDV1-VILP and IGF-1 are of similar length (10 vs 12 amino acids) and share 33% sequence identity, including Arg37, one of the residues in the IGF-1 C-domain known to be in the receptor binding interaction[41]. Removal of the C-peptide from LCDV1-VILP decreases the affinity of LCDV1-VILP for the IGF1R by tenfold, consistent with the important role for the C-peptide in the receptor binding, similar to what is observed with IGF-1 itself[37,42,43]. scLCDV1-VILPs exhibits very low affinity for binding to the IR and activating IR post-receptor signaling (about 1/1000 as active as insulin), and this is not increased with dc-LCDV1-VILP, despite having a more insulin-like structure. Consistent with this, glycemia in mice expressing LCDV1-VILP in liver was not changed compared to controls.

LCDV1-VILP does have one unique and important property, namely its ability to antagonize IGF-1 at the IGF1R. This effect of LCDV1-VILP is restricted to the single chain peptide and is seen only with the IGF1R, and not with the IR. The fact that scLCDV1-VILP has the potential to serve as an IGF-1 receptor antagonist is unique among naturally occurring insulins and IGFs. Furthermore, our data show that the potency of scLCDV1-VILP to inhibit IGF1R signaling is sufficient to have functional consequences in vitro and in vivo. Thus, scLCDV1-VILP is able to reduce IGF-1 induced cell proliferation in vitro and partially block the effects of endogenous IGF-1 in transgenic mice over-expressing bovine growth hormone in vivo, as manifested by reduced lean body mass and reduced body weight gain. Since this antagonism is restricted to the single chain version of the hormone, this suggests that when over-expressed in liver, LCDV1-VILP is not processed, but secreted as a single chain molecule, similar to IGF-1.

Consistent with the current study, our groups have shown that LCDV1-VILP can also serve as an inhibitor of IGF1R in HEK293 cells, and using chimeric peptides of IGF-1 and LCDV1-VILP, shown that the C-peptide domain of the VILP is essential for its antagonistic action[29]. In addition, one amino acid, a glycine in position 7 in the B-domain of IGF-1 also appears to be required for the antagonist effects. This glycine is highly conserved in most members of the insulin/IGF-1 superfamily[44] and is replaced by a serine in LCDV1-VILP. This residue also appears to be required for the antagonist effects, since substitution of the serine by a glycine decreases the potency of LCDV1-VILP to inhibit IGF-1 signaling and increases its agonist property[29]. Also of interest is the finding that, despite its low agonist activity, scLCDV1-VILP has unique post-receptor signaling properties, such that at a concentration of 10 nM, it can activate the AKT pathway, but does not appear to activate the ERK pathway. Since scLCDV1-VILP is capable of inducing IGF1R internalization similar to IGF-1 itself, this selective signaling seems to be independent of IGF1R endocytosis. Which structural features account for this biased signaling remain to be determined.

In addition to demonstrating its unique signaling and biological activities in vitro and in vivo, in the present study we were also able to determine the unique nature of the LCDV1-VILP receptor interaction at a structural level using cryoEM. While previous studies have shown that IGF1R binds only a single IGF-1 or IGF-2 molecule[35,37], even when prepared with molar excesses of these ligands, we find that the IGF1R is able to bind two molecules of scLCDV1-VILP. This two-ligand-bound structure appears to arise as a consequence of the [L1-CR + αCT ́]-bound scLCDV1-VILP being unable to engage productively with site 1b. We propose that when a single scLCDV1-VILP binds to the IGF1R, the

ligand-bound L1-CR module detaches from the FnIII-2 ́ domain, partially freeing up the three-domain FnIII ́ module and permitting it to move into proximity of the FnIII domain module in the opposing monomer. As the fully stabilized conformation of an IGF-1-liganded ectodomain is not achieved, such movement may reduce agonism compared to the IGF-1 occupied receptor ("partial agonism"). Also, in the absence of the conformational transition that enables the bound scLCDV1-VILP to engage site 1b, a second scLCDV1-VILP can bind to the alternate [L1-CR] ́ + αCT element, yielding the double-liganded structure observed in our study. The overall conformational flexibility of the ectodomain is also reduced by the constraints imposed by the disulfide-linked αCT elements that cross-link the two scLCDV1-VILP-bound L1-CR modules, resulting in the ectodomain's points of membrane entry and the associated intracellular domains becoming further separated, rather than closer together as they do when IGF-1 binds. As a result, transphosphorylation of the intracellular beta-subunits does not occur, and the receptor is unable to activate its downstream signaling pathways.

The potency of scLCDV1-VILP to activate IGF1R and its downstream signaling may vary depending on the concentration of ligand. Whereas higher concentrations (100–1000 nM) of scLCDV1-VILP are associated with the inactivation of the receptor, lower levels (1–10 nM) of ligand could potentially lead to partial IGF1R activation. Since the cryoEM was performed in the presence of a large excess of the ligand, we hypothesize that at low concentrations, it may be possible for one scLCDV1-VILP to bind to the receptor and induce a conformational change leading to the partial activation of the receptor, further studies will be required to know if this actually occurs.

As noted above, studies of chimeric and mutant IGFs show that the antagonistic properties of scLCDV1-VILP are associated with its unique C-peptide sequence and with a serine at position 8 in the B-domain (equivalent to a glycine in IGF-1 at position 7)[29]. Within the AF2 predicted structure, scLCDV1-VILP Ser8 lies at the N-terminus of the B-domain helix in scLCDV1-VILP which would, compared to glycine, impose a Ramachandran constraint on the positioning of residues B1-B7. Within our model, Ser8 could interact with the nearby residues Glu91 and Lys115 within domain L1 and Glu693 ́ within αCT ́, though we acknowledge the resolution of the cryoEM map precludes confirmation. These differences may impact the ability of scLCDV1-VILP to engage key Site 1b residues, Lys530, Tyr487 and Arg488 from FnIII-1 ́[37] that compete for the same intra-receptor interactions. Differences in the structure of the respective C domains may also contribute to the unique interaction. Replacement of the scLCDV1-VILP C-domain residue Arg32 by its critical hIGF-1 counterpart Tyr31 does not reduce antagonism, but replacement of the scLCDV1-VILP Arg31-Arg32-Arg33 triplet by its Gly-Tyr-Gly IGF-1 equivalent does[29]. In our structure, the C domain of scLCDV1-VILP appears relatively displaced towards IGF1R domain L1, which may hinder it forming a native-like interaction with domain L2. Moreover, the residue Glu9 in the B-chain of IGF-1 forms a salt bridge with the Tyr448 of the site 1b[35,37]. Glu9 is highly conserved in insulin superfamily members. However, this residue is replaced by a histidine in LCDV1-VILP. The absence of this salt bridge may also reduce affinity for the Site 1b.

IGFs and IGF signaling have a key role in growth and cell proliferation, and deregulation of these processes contributes to the pathogenesis of multiple diseases[45]. Overproduction of IGF-1 can lead to excessive growth in patients with acromegaly or gigantism[46], and high expression of IGF-1 can also promote growth of many types of tumors[47]. Excessive secretion of incompletely processed IGF-2 has been shown to be a cause of non-islet cell tumor hypoglycemia[48]. Overexpression of IGF1R has also been shown to promote a number of cancers, including breast, gastrointestinal and prostate cancer[49]. Conversely, inhibitors of IGF1R have been shown to have beneficial effects in some cancers[50], alleviate the inflammatory process associated with diabetic kidney disease in mice[51], and are already in use in

humans in the treatment of hyperthyroidism associated eye disease[52]. Thus, developing inhibitors of IGF1R signaling could provide important therapeutics. Because of the strong homology between IGF1R and IR, the development of IGF1R inhibitors has been challenging and most did not achieve their therapeutic goals due to a lack of specificity and/ or only modest efficacy[53]. scLCDV1-VILP is the first natural peptide that exhibits a specific antagonist potency on IGF1R. Thus, this peptide or peptides based on its structure may serve as a guide for development of IGF1 antagonists.

## Methods

### Bioinformatics

The sequence of LCDV1-VILP was compared to those of insulin and IGF-1 by performing a multiple sequence comparison by log-expectation (MUSCLE)[54] using Molecular Evolutionary Genetics Analysis software (MEGA version 10.1.8). The whole LCDV1-VILP sequence was used on BlastP to identify the top 10 sequences that exhibit the highest identity. AlphaFold2 was used to predict three-dimensional structures of scLCDV1-VILP and dcLCDV1-VILP[36]. Structures were visualized using ChimeraX[55].

### Peptide synthesis

**Single chain LCDV1-VILP**. Single chain viral insulins were synthesized on 0.1 mmol Rink amide ChemMatrix® (PCAS BioMatrix Inc.) resin using an ABI-433A peptide synthesizer and Fmoc/6-Cl-HOBt/DIC coupling protocols[29]. Fmoc-Asp-OtBu was employed to introduce the C-terminal Asn. Cleavage was conducted by treatment with 10 mL of a TFA solution containing 2.5% TIS, 2.5% 2-mercaptoethanol, 2.5% anisole, and 2.5% $H_2O$ at room temperature with gentle agitation for 1.5 h. The resin was filtered, and the peptide precipitated by addition of cold ether (50 mL). The peptide precipitate was collected by centrifugation then washed with cold ether (3 × 50 mL). The crude peptide was solubilized in mixture of ammonium acetate (0.0429 M), urea (2.571 M) and acetic acid (0.01 M). $I_2$ in methanol was added dropwise, and the solution was further stirred for 20 min before quenching with ascorbic acid. The intermediate with two disulfide bonds was obtained after preparative HPLC purification and lyophilization. The lyophilized peptide was dissolved in TFA with the addition of DMSO (5%). The mixture was stirred at room temperature for 2 h. After cold ether precipitation, the peptide was solubilized in 20% $CH_3CN/H_2O$ and adjusted to pH 8.0. The solution was acidified after 30 min at room temperature. Another preparative HPLC purification and lyophilization to yield the final product.

**Double chain LCDV1-VILP**. The A-chain was assembled on Rink amide ChemMatrix resin with Fmoc-Asp-OtBu to introduce the C-terminal Asn. The B-chain was constructed on Thr-preloaded HMPB ChemMatrix resin. Both peptides were cleaved and purified as described previously[29]. A-chain (0.0048 mmol) and B-chain (0.0048 mmol) were then mixed in 2.8 mL of ammonium acetate (0.0143 M) and urea (0.0857 M, pH 6.9). The thiolysis chain combination was monitored by LC-MS analysis. Upon completion, the mixture was diluted with acetic acid (0.01 M) and $H_2O$. $I_2$ (30 mg) in 0.8 mL methanol was added, and the solution was stirred for 20 min. Aqueous ascorbic acid solution was added to quench the reaction to yield a colorless solution. The resulting dimer was purified with preparative HPLC. The A-B heterodimer with two Cys(tBu) residues was dissolved in TFA (1.9 mL) followed by addition of DMSO (0.1 mL). The mixture was stirred at room temperature for 2 h. The crude product was recovered by ether precipitation. The final product was obtained after prep-HPLC and lyophilization.

### Competition binding assay

Stable CHO Lec8 cells line expressing respectively the IGF1R ectodomain (Δβ mutant) and the IR-A ectodomain were obtained from CSIRO (Parkville, Australia), with the protein being expressed and purified as previously described[56,57]. Competitive binding assays was then performed as previously described with slight modification[20]. Briefly, wells of a 96-well plate were coated with 250 ng of an IGF1R monoclonal antibody (24-31) or an IR monoclonal antibody (83-7), gifts from Professor Ken Siddle (University of Cambridge). After an overnight incubation at 4 °C, the antibody solution was removed, and the wells were blocked with 0.5 % bovine serum albumin (BSA) in 20 nM Tris pH 7.4, 150 nM NaCl and 0.1% tween-20 (TBS-T) for 2 h at room temperature. Ectodomain protein was then added for 2–3 h at room temperature in TBS-T (0.5 µg/well). $I^{125}$ labeled IGF-1 or insulin (0.01 nM) was added in each well, in addition of an increasing concentrations of unlabeled IGF-1, insulin, or LCDV1-VILPs for 16 h at 4 °C. Wells were washed three times with TBS-T, and 100 µL of a solution containing NaOH (0.1 nM) and SDS (0.1%) was added. After 10 min, 80 µL of the lysis solution was added to a 4 mL of scintillation liquid (Perkin Elmer), and the radioactivity was measured in a liquid scintillation counter (Beckman).

### IR and IGF1R signaling

MCF-7 cells, a breast cancer cell line (SL017, Genecopoeia) and brown preadipocytes knocked-out for the endogenous insulin and IGF-1 receptors and overexpressing either the human IR or the human IGF-1 receptor[31] were plated into six-well plates. Cells were cultured in DMEM supplemented with 10% FBS (Sigma), 100 U/mL penicillin, 100 µg/mL streptomycin (Gibco) and 100 µg/mL Normocin (Invivogen) at 37 C with 5% $CO_2$ and 90% humidity. When cells reached 70–80% confluence, the medium was removed and replaced by a starvation medium, i.e., without FBS, for 6 h. Cells were then incubated for 15 min with insulin, IGF-1 or LCDV1-VILP at concentrations from 0.1 to 1000 nM. In the case of co-incubation experiments, cells were pre-treated with the LCDV1-VILP for 30 min and then stimulated with 10 nM insulin or IGF-1 for 15 min. Cells were washed with ice cold PBS and snap frozen in liquid nitrogen. Proteins were extracted with a RIPA buffer (MilliporeSigma) supplemented with 0.1% SDS and a cocktail of protein phosphatase (#B15001-A and B15001-B, Bimake) and protease inhibitors (#B14001, Bimake) at 1x. Protein concentrations were assessed using a BCA protein assay kit (Pierce). Proteins (10–20 µg) were loaded in a 4–12% Bis-Tris gel (Invitrogen) and transferred to a PVDF membrane. The membrane was incubated at room temperature for 1 h in a block solution (ThermoFisher) before an overnight incubation at 4 °C with primary antibodies (1:1000). Membranes were washed in a TBS-T buffer then incubated for 4 h at room temperature with HRP conjugated secondary antibodies (1:1000). The following antibodies were used in this study: IRS-1 (catalog 611394) from BD Bioscience, p-IRS-1$^{Y895}$ (catalog 3070), IR (catalog 3025), IGF1R (catalog 3027), p-IR$^{Y1150/1151}$/IGF1R$^{Y1135/1136}$ (catalog 3024), AKT (catalog 4685), p-AKT$^{Ser473}$ (catalog 4060), ERK1/2 (catalog 9102) and p-ERK1/2$^{T202/Y204}$ (catalog 4370) from CellSignaling, goat anti-rabbit HRP conjugated (catalog 1706515) from BioRad and sheep anti-mouse HRP conjugated (catalog NA931) from Sigma-Millipore. Quantification of protein bands intensity were analyzed on ImageJ (v2.3.0/1.53q) and Microsoft Excel (v16.64). All the uncropped and unprocessed pictures of each blot are available in the source data file.

### IGF1R internalization assay

Murine brown preadipocytes overexpressing hIGF1R were seeded into six-well plates. When cells reached 90% of confluence the complete medium was removed and replaced by serum-free medium for 3 h. Cells where then incubated for 45 min with IGF1 or scLCDV1-VILP at 10 or 1000 nM. Cells were washed with ice-cold PBS (pH = 8) and labeled with 0.3 mg/ml of sulfo-NHS-biotin (#21335, ThermoFischer) for 30 min at 4 °C. The labeling solution was removed and replaced by an ice-cold solution of 100 mM of glycine for 10 min at 4 °C.Cells were washed two times with ice cold PBS and lysed in a RIPA buffer containing a cocktail of protein phosphatase (#B15001-A and B15001-B,

Bimake) and protease inhibitors (#B14001, Bimake) at 1x. Protein concentrations were assessed using a BCA protein assay kit (Pierce). Biotinylated membrane proteins were enriched by incubating 15 µl of streptavidin-agarose beads solution (#20357, ThermoFisher) with 120 µg of protein lysate in 800 µL final volume overnight at 4 °C.Agarose beads were pelleted by centrifugation (5 min, 9000 × g, 4 °C) and washed three times in RIPA lysis buffer. Bound proteins were detached from agarose beads by heating for 5 min at 95 °C in 1× loading buffer. Biotinylated membrane proteins and total proteins were loaded in a 4-12% Bis-Tris gel (Invitrogen) and transferred to a PVDF membrane. Membranes were immunoblotted with the following antibody: p-IR[Y1150/1151]/IGF1R[Y1135/1136] (catalog 3024), IGF1R (catalog 3027) from CellSignaling and goat anti-rabbit HRP conjugated (catalog 1706515) from BioRad.

### Proliferation and survival assays

Proliferation and survival assays were performed on murine brown preadipocyte overexpressing the human IGF1R[31]. For the proliferation assay, cells were seeded on a 48-well plate (20,000 cells/well) and incubated with IGF-1 (Peprotech), scLCDV1-VILP and dcLCDV1-VILP at concentrations from 0.1 to 1000 nM. After a 48-h incubation at 37°C, cells were counted using a cell counter (Cellometer). For the survival assay, cells were seeded into a 96-well plate. Once they reached 100% confluence, the cells were starved overnight. The cells were incubated with 1000 nM of scLCDV1-VILP or dcLCDV1-VILP. After 48 h, the cells were washed with PBS and stained with a 0.5% solution of crystal violet (Sigma) in 20% methanol for 10 min at room temperature. Each well was washed three times with PBS, and the cells were dissolved in 100 µL of methanol. Absorbance was read at 560 nm.

### Quantitative reverse transcription PCR

Messenger RNAs were extracted from liver biopsies using a mix of TRIzol® reagent (Invitrogen) and chloroform and then precipitated with 70% of ethanol. Messenger RNA purification was performed using RNeasy Mini Kit columns (QIAGEN) according to the manufacturer's protocol. Complementary DNA were synthesized by reverse transcription of 2 µg mRNA using a high-capacity cDNA synthesis kit (Applied Biosystems). The qPCR was performed using a Sybr Green Supermix (Biorad) on a CFX384 real time PCR detection system (Biorad). The following primers were used: *Igf-1* forward: 5′-GTGGATGCTCTT-CAGTTCGTGTG-3′, *Igf-1* reverse: 5′-TCCAGTCTCCTCAGATCACAGC-3′, *Lcdv1-vilp* forward: 5′AGAAGAAGCACCAG-AAACGG-3′ and *Lcdv1-vilp* reverse: 5′-TTTCCAGGTCGTCGGTTGTAC-3′.

### Recombinant AAV vectors

The AAV expression cassette was obtained by cloning, between the inverted terminal repeats (ITRs) of AAV2, the coding sequence of LCDV1-VILP under the control of the liver-specific human α1-antitrypsin promoter (hAAT). A non-coding cassette carrying the hAAT promoter, but no transgene, was used to produce control null vectors (AAV-null). Single-stranded AAV vectors of serotype 8 (AAV8) were produced by triple transfection in HEK293 cells and purified using an optimized CsCl gradient-based purification protocol that renders vector preps of high purity and devoid of empty capsids[58]. Viral genome titers were determined by Picogreen using phage lambda DNA as standard curve.

### bGH mice

Bovine growth hormone transgenic mice were generated and identified as previously described[59]. Briefly, the bGH were generated using a metallothionein transcriptional regulatory element linked to the first exon and intron of the bGH cDNA. This construct was injected into the pronucleus of C57BL/6J embryos. Mice containing the transgene were identified with PCR analysis of DNA from tail biopsy specimens obtained 4 weeks after birth. The primers used for genotyping are the same as described previously for bovine GH antagonist transgenic mice[60]. Nineteen-day old female mice were injected via the tail vein with the AAV8 construct above designed to overexpress LCDV1 in liver. Each mouse received $1 \times 10^{12}$ viral genomes (vg) of either the AAV8 vectors carrying the LCDV1-VILP sequence (n = 8) or an AAV8 control (n = 6) that only carry the hAAT promoter and a poly (A) sequence. Body composition was assessed weekly using the Bruker Minispec (Woodlands, TX), which uses NMR technology to estimate the fat and lean mass of the animals[33]. Tail vein blood glucose was measured 2 and 5 weeks after the AAV injection using Onetouch Ultra test strips and glucometers (Lifescan, Inc, Milpitas, California). Mice were euthanized 8 weeks after the AAV injection, body length was measured and tissues (liver, skeletal muscles, brown and white adipose tissues) were collected, weighed, snap frozen and stored at −80 °C until further analysis. Mice were housed 2–4 per cage and given ad libitum access to water and rodent chow (ProLab RMH 3000; 14% of energy from fat, 60% from carbohydrates, and 26% from protein; PMI Nutrition International, Brentwood, Missouri). The cages were maintained in a temperature (22 °C) and humidity-controlled room and exposed to a 14-h light, 10-h dark cycle. All procedures were approved by the Ohio University Institutional Animal Care and Use Committee and fully complied with all federal, state, and local policies.

### Cloning and production of IGF1Rzip

The construct IGF1Rzip comprises a 30-residue signal peptide, followed in order by residues 1–905 of the intact IGF1 holoreceptor, a 33-residue leucine-zipper motif[61], a three-residue spacer t, and an eleven-residue c-myc tag. The leucine-zipper motif is proposed to act as a "soft restraint" on the spatial separation of the FnIII-3 domains and to mimic membrane embedding[35,62,63]. Briefly, the construct was produced by stable expression and secretion from CHO-K1 cells and then purified by a sequential combination of 9E10 antibody-affinity chromatography and size-exclusion chromatography as previously described[35], with the final sample being prepared at a concentration of 0.2 mg mL$^{-1}$ in a 24.8 mM Tris-HCl (pH 8.0), 137 mM NaCl, 2.7 mM KCl plus 0.02% NaN$_3$ ("TBSA").

### CryoEM sample preparation

scLCDV1-VILP was prepared at a concentration of 150 µM in 10 mM HCl (=1 mg mL$^{-1}$) then diluted 2:5 (v/v) in TBSA (described above) to give a 60 µM solution in TBSA plus 4 mM HCl. The resultant solution was the combined 1:1 with IGF1Rzip sample prepared as above to provide a final sample concentration of ~0.1 mg mL$^{-1}$ IGF1Rzip and 30 µM scLCDV1-VILP in TBSA plus 2 mM HCl (~30-fold molar excess calculated per receptor αβ monomer).

### CryoEM imaging

Quantifoil Cu 1.2/1.3 grids (200 mesh) were glow discharged using a 30 mA current for 30 s. The sample was then applied to the glow-discharged grids and blotted to achieve desired sample film thickness (Thermo Fisher Scientific Vitrobot; settings: 3 s blot time, -3 blot force, 100% humidity, 4 °C) before freeze plunging. Movie data were collected using a K2 camera (Gatan) attached to a Titan Krios cryo-electron microscope (Thermo Fisher Scientific) operating at 300 kV (C2 aperture: 50 mm; magnification: 130,000×; energy-filtered zero-loss imaging mode with slit width 20 eV; pixel size: 1.06 Å; total exposure 50 e$^-$ Å$^{-2}$; 50 frames per movie; dose rate: 6 e$^-$ px$^{-1}$ s$^{-1}$) (Supplementary Fig. 5a–c).

### CryoEM 3D reconstruction

3025 movies underwent motion-correction and dose-weighting using the patch-motion correction job within cryoSPARC v3.2[64], followed by CTF parameter estimation using the patchCTF job (Supplementary Fig. 6). Micrograph quality was then assessed, with only micrographs comprising thin vitreous ice, CTF fit better than 4.3 Å and low motion

curvature retained, giving a set of 2426 good micrographs. Particle coordinates were determined using crYOLO 1.8.0-beta[65], giving 1.36 M particles, which were re-imported, binned 4× and extracted with a 64 px box size. The binned particles were 2D classified into 200 classes, and 410k particles from 45 good 2D classes were retained. Junk particles were removed by heterogeneous refinement into two classes, with a noisy initial model employed as a sink for poor particles. 255k particles corresponding to the good class were extracted at full resolution with a box size of 360 px, and 3D classified into three classes using the ab initio reconstruction job. The most complete class, containing 89k particles, was then reconstructed using non-uniform refinement to give a nominally 4.6 Å resolution map. All classification and reconstruction steps were carried out within cryoSPARC v3.2. Further processing (viz., 3D classification in cryoSPARC or Relion 3, per-micrograph and per-particle CTF refinement and local refinements) did not result in improved reconstructions.

### Model building

An atomic model of the scLCDV1-VILP-bound IGF1Rzip was built and refined as follows. First, individual copies of receptor modules L1-CR, L2, FnIII-1, and FnIII-2 were extracted from PDB entry 5U8R and docked manually into the map using ChimeraX[55]. The model was then rigid-body fit using Phenix version 1.19.2-4158-000[66], treating each of the above structural modules as individual rigid entities. Two copies of the extended αCT segment (extracted from PDB entry 6PHY) and of the scLCDV1-VILP moieties (generated with AlphaFold2[36]) were then added to the model based on their discerned location within the map. Manual editing with secondary structure, torsion and Ramachandran restraints was undertaken using Coot version 0.9[67] to regularize inter-domain junctions and to generate a plausible conformation for the C peptide of each scLCDV1-VILP. A final rigid-body fit with the two receptor monomers and two scLCDV1-VILP molecules as four rigid entities was then undertaken. Refinement statistics are in supplementary Table 1. Map density in Fig. 5a, b was generated from the final reconstruction using the 'volume zone' tool in ChimeraX 1.3 with an atomic distance of 2.5 Å. Map vs model FSC curve can be found in supplementary Fig. 4f.

### Reporting summary

Further information on research design is available in the Nature Research Reporting Summary linked to this article.

## Data availability

CryoEM map and atomic model of scLCDV1-VILP bound to IGF1R.zip generated in this study have been deposited in the EMDB and PDB databases, respectively, under the accession codes EMD-26306 and 7U23. The modules L1-CR, L2, FnIII-1, and FnIII-2 were extracted from PDB entry 5U8R. The two copies of the extended αCT segment were extracted from PDB entry 6PHY. Source data are provided with this paper.

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

## Acknowledgements

This work was supported by the National Institute of Diabetes and Digestive and Kidney Diseases of the National Institutes of Health under award number R01DK031036 (to C.R.K.) and K01DK117967 (to E.A.). We thank Ms Mai Margetts and Dr Yibin Xu (WEHI, Parkville) for the provision of an aliquot of IGF1Rzip for use in these studies and Prof. Ken Siddle (University of Cambridge) for providing MCL's laboratory with the hybridomas used to express mAb 26-60 and mAb 83-7. Stable cell lines expressing receptor ectodomains were obtained from CSIRO (Parkville, Australia) by WEHI under a Technology Transfer Agreement. WEHI receives Victorian State Government Operational Infrastructure Support and funding from the Australian NHMRC Independent Research Institutes Infrastructure Support Scheme. The authors acknowledge the use of instruments and assistance at the Monash Ramaciotti Centre for Cryo-

 

Electron Microscopy, a Node of Microscopy Australia and the use of equipment funded by the Australian Research Council grant LE120100090.

## Author contributions
F.M., E.A., and C.R.K. design the study, F.M. and M.C. carried out the experiments, F.M. and C.R.K. analyzed the data, F.M. and C.R.K. wrote the manuscript with the help of M.C.L. and N.S.K. for the part dedicated to the cryoEM experiment., F.Z., V.G., and R.D.D. purified the VILPs, E.O.L. and J.J.K. undertook experiments on BGH mice, V.J. and F.B. produced the AAV8, N.S.K. and H.V. undertook cryoEM experiments, N.S.K. undertook cryoEM reconstruction, M.C.L. and N.S.K. undertook molecular model building, C.R.K. supervised the study. F.M. and N.S.K. contributed equally to this manuscript. All authors discussed the data and contributed to the final manuscript.

## Competing interests
M.C.L.'s laboratory has a funded agreement with Eli Lilly and Company to conduct research not connected to this publication. The remaining authors declare no competing interests.
