## [Peer Review File · Nature Communications]

Interaction of a Viral Insulin-Like Peptide with the IGF-1 Receptor Produces A Natural AntagonistReviewer #1 (Remarks to the Author):

NCOMMS_22_11451

Moreau et al present data on a viral insulin-like peptide (VILP) from Lymphocystis disease virus-1 (LCDV1) with high affinity for human and murine IGF1R. The authors show data indicating that LCDV1-VILP has unique antagonistic actions for IGF1R in vitro and in vivo and used cryo-EM to reveal its structural receptor binding interactions. Much of the in vitro signaling has been previously demonstrated by a prior publication (Altindis et al, PNAS 115:2461, 2018), and some of the current findings contradict what the group found previously in terms of IR signaling and preferential stimulation of the AKT compared to ERK pathways after binding to IGF1R. The in vivo findings in the current study show that treatment of mice expressing hGH with LCDV1-VILP through AAV delivery reduces body weight without affecting blood glucose levels indicating in vivo antagonist actions. This is in contrast to their prior findings that acute injection of the peptide into mice caused a significant reduction in blood glucose (Altindis, 2018). Moreover, another publication by some of the same authors demonstrated that LCDV1-VILP has antagonistic properties for the IGF1R through its C-peptide domain (Zhang et al. Mol. Metab. 53:101316, 2021). The major new contribution in this study is in the cryo-EM structural analyses of the peptide and IGF1R binding. The value of the studies on this peptide is in the potential for understanding how to use it to develop IGF1 antagonists for use in patients.

Major Comments

- 1. Figure 2, panels a,b – for completeness, it is important to include the competition binding assays using the B isoform of the hIR as well as panels c-f shown for the A isoform. There is no explanation why this Figure shows mixed results for the A and B isoforms. Is it entirely clear they will be the same? Moreover, based on Altindis, 2018 it is clear the A isoform may show stronger binding.**
- 2. As the authors point out, it is interesting that the LCDV1-VILP activated Akt but showed no stimulation of ERK phosphorylation whereas IGF-1 and insulin activated both pathways. The conclusion presented is that there may be “a unique post-receptor pathway bias in the effects of these ligands”. This raises the question of whether binding of the LCDV1-VILP ligands do not result in receptor internalization and ERK signaling in endosomes.**
- 3. Figure 3 – It is important to do this experiment in human cells expressing endogenous levels of both the IGF1R and IR. What might be the outcome based on no overexpression or on having some hybrid receptors present? Some of these studies (on other cell types) are presented in the prior Altindis publication with conflicting results to those presented here. This needs to be addressed.**
- 4. Figure 4a,b – Cell number after 48 hr treatment was used to quantify proliferation. This could also be altered by changes in survival that should be included. Alternatively, a direct measure of proliferation is needed.**
- 5. In vivo studies on the hGH transgenic mice:**
 - a) There is no discussion of why only female mice were used for the in vivo studies on the hGH transgenic mice. Was the effect on male mice similar?**
 - b) In these experiments, the authors include an AAV only control. Another control should be included for growth measurements to show that AAV-IGF1 itself does not have the same effect on body growth as the AAV-LCDV1.**
- 6. A major conclusion from the structural analyses is that the LCDV1-VILP ligand binding to the IGF1R causes an inactive conformational state such that it cannot activate downstream signaling. It is then unclear how Akt is activated as shown in Figure 2.**

Minor Comments

Figure 5

a) Panels a,b – the models presented here are based on prior publications. References are provided but it is not clear if the images have been reconstructed or reproduced here with permission.

b) Panels g,h – no dashed line is apparent in the figure as indicated by the legend.

Reviewer #2 (Remarks to the Author):

The manuscript by Moreau et al. characterizes single- and double-chain viral insulin-like peptides (VILPs) encoded by Lymphocystis disease virus-1 (LCDV-1). The authors show through a series of biochemical and in vivo mouse experiments that the single-chain version is a potent antagonist of insulin-like growth factor-1 receptor (IGF1R), with no or minimal effect on signaling through the insulin receptor (IR).

They then characterize the binding of scLCDV1-VILP to the ectodomain of IGF1R using cryo-electron microscopy. In contrast to the 1:1 complex observed previously between IGF1 and IGF1R, the authors find two molecules of scLCDV1-VILP bound to IGF1R in the two 1a binding sites. The peptides do not engage the two 1b sites, which has the effect of increasing the distance between the ends of FnIII-3 to 90 Å from 67 Å for the apo IGF1R. This will keep the cytoplasmic kinase domains from undergoing trans-phosphorylation, explaining how scLCDV1-VILP acts as an IGF1R antagonist.

This study presents findings that will be of great interest to those in the insulin/IGF1 signaling field.

Specific point:

One issue related to the 2:1 stoichiometry of VILP peptide to IGF1R is the 30-fold molar excess of peptide used in the cryo-EM study. How do we know that the 2:1 stoichiometry, rather than 1:1, is the relevant state in vivo?

Reviewer #3 (Remarks to the Author):

Moreau and Kirk et al. have identified a viral insulin-like peptide that can act as an antagonist of IGF-1R with no/limited effects on the highly homologous insulin receptor. The cryo-EM structure reveals a novel 2:1 ligand:IGF-1R complex that explains the broad mechanism by which scLCDV1-VILP acts as an antagonist. In-vivo experiments in mice show that expression of LCDV1-VILP causes a reversal of an IGF-1R overstimulation phenotype without adverse effects on resting blood glucose, reaffirming the therapeutic utility of an IGF-1R antagonist. This work advances our basic understanding of how IGF-1R is activated and offers valuable insight towards the development of potential IGF-1R antagonists that have high selectivity for IGF-1R vs IR.

Suggested revisions:

1. Only one of the citations on line 65 has results that show the membrane proximal spacing. Ref 20 is only the first three domains of IGF-1R.
2. On line 119, the sequence of scLCDV1-VILP and dcLCDV1-VILP are the same and therefore the dc/sc prefix should be removed
3. On line 66, the 67 Å spacing applies only to IGF-1R. The spacing for IR is closer to 115 Å. The wording could be changed from "these receptors" to "the IGF-1 receptor".
4. Figure 3B and line 179: While the effect is not statistically significant because of the large range in measured values, the average fold-change for IR-induced ERK phosphorylation (3d, right) is of similar magnitude to the ERK suppression seen in the IGF-1R case (3c right). I suggest the authors collect additional data to improve

statistical power or soften the language here. Another way the authors could gain supporting data for this idea is to perform the cell growth assay in cells over-expressing IR in addition to the data already shown for cells overexpressing IGF-1R.

5. Please add units for the y-axes in Figure 3.

6. On line 186, the phrase "the curve was shifted two orders of magnitude to the right" does not seem consistent with Fig. 4a. (1 nM to 10 nM is one order of magnitude)

7. For Fig 4c, the magnitude of the weight loss effect is not apparent without reading about these mice. Would it make sense to show the weight of the control mice over the same period?

8. Line 281: the wording "binds as a monomer" is confusing as both IGF-I and scLCDV1-VILP bind to IGF-1R as monomers.

9. On line 366, mention of the Gly to Ser mutation in the C-peptide is the first mention within the paper. Should it say B chain instead of C-peptide?

10. I am glad the authors mentioned the C-peptide context being somewhat different at site1a and site 1b. This could explain the importance of the VILP C-peptide sequence for antagonism. If there are any interactions with the receptor that are expected to be different in the site 1a and site 1b states, I suggest making a supplementary figure illustrating this; however, it is not clear to me that there are such interactions apparent from the cryo-EM data without seeing the atomic coordinates, so I leave this to the discretion of the authors.

Reviewer #4 (Remarks to the Author):

Moreau, Kirk, et al. characterize two viral insulin/IGF-1 like peptides (VILPs) using various in vitro and in vivo methods.

Using a competition binding assay they show that the single chain peptide (scLCDV1-VILP) is more potent in competing with Insulin and IGF-1 for binding their respective receptor compared to the double chain peptide (dcLCDV1-VILP). However, all determined VILP binding affinities are lower than the ones of insulin and IGF-1 to their respective receptor. scLCDV1-VILP showed the highest affinity to hIGF1R still being an order of magnitude lower than that of IGF-1 itself.

In line with these findings only scLCDV1-VILP showed a notable effect on IGF1R signaling in cell culture experiments. The authors went on to show that scLCDV1-VILP has an antagonistic effect on IGF-1 induced IGF1R signaling while it had no effect on insulin induced IR signaling.

Consequently, they show that scLCDV1-VILP does inhibit IGF-1 induced cell proliferation and mouse growth.

In order to explain this antagonistic effect on a molecular level, the authors turned to cryo-EM to determine the structure of scLCDV1-VILP in complex with the ectodomain of IGF1R. Using single particle analysis they solved the structure at a nominal resolution of 4.6 angstrom and derived an atomic model thereof. The presented structure, however, is not convincing to back up the drawn conclusions and does not live up to community standards. Therefore I unfortunately cannot recommend publication of the manuscript in the journal Nature Communications.

Major issues

1) line 208ff and Figure 5: the novel findings from this study should be emphasized more, i.e. it should be made more clear that the model shown in panel c is the new finding. The reconstructed cryo-EM density also has to be presented in the main text and figures and not only the last supplementary figure S4 with only one view. The chosen surface representation of the derived model is misleading in regards to the claimed resolution of the cryo-EM reconstruction. In the main text, previously published structures are discussed at length before the new findings from this study are presented relatively brief. It should be the other way round.

2) Figure S4: The relatively flat fall-off of the FSC starting from low frequencies points at

issues in the particle alignment and 3D reconstruction. With such 'pathogenic' FSC curves the nominal resolution as given by the interception at 0.143 has to be taken with care and the resolution claim has to be backed up by the presence of characteristic features. At 4.6 Å, alpha-helices including the pitch have to be clearly visible. Another characteristic feature in this resolution range is the beginning separation of beta-strands in the density. Neither of those features can be observed in the shown density map, and if present, have to be presented in a readily accessible way. In order to improve the 3D reconstruction, the authors could employ 3D classification in Relion as Ab-initio reconstruction as implemented in cryoSPARC is not always suited well for classifying particles.

3) line 604ff: Even though atomic coordinate model building is technically possible at 4.6 Å resolution, it is a very challenging task and requires a high-quality reconstruction. With the above mentioned limitations regarding map quality, I don't believe atomic coordinate model building is appropriate in this case. This is also evident from the poor atom inclusion shown on page 6 and 7 of the wwPDB EM Validation Report. The authors should limit the map interpretation to rigid-body fitting of known parts of the structure. In case atomic coordinate model building is pursued, at least a map-to-model FSC should be presented. A cut-off value is reported in Table 2, but it is not evident where this value comes from. For map-to-model FSCs, the reported cutoff should also be at 0.5 instead of 0.143.

4) Figure S2: the detailed structural analysis on the side-chain level, especially in panel d and e should be avoided as it gives the reader a false sense of model quality. In the presented cryo-EM density basically no side chain densities are visible.

Minor issues

*Figure 1c: it should be pointed out in the figure or the legend that the presented models for the VILPs represent AlphaFold2 predictions. A representation of the AlphaFold2 confident score (pLDDT) should also be included. PDB codes of the shown insulin and IGF-1 models should be indicated.

*Figure 4d: the Y-axis numbering seems to be partially covered by the label.

*Figure 5a/b: PDB codes of the already published structures should be indicated

*Figure S3a: scale bar is missing from the micrograph

*line 233: further refinement focusing on [...]: it is not clear from the methods section what is meant by that statement. Which density is that statement referring to? If it's not shown in the supplement yet, it should be included.

*line 608: what is meant by 'rigid body real-space refinement'?

REVIEWER COMMENTS

Reviewer #1 (Remarks to the Author):

Moreau et al present data on a viral insulin-like peptide (VILP) from Lymphocystis disease virus-1 (LCDV1) with high affinity for human and murine IGF1R. The authors show data indicating that LCDV1-VILP has unique antagonistic actions for IGF1R in vitro and in vivo and used cryo-EM to reveal its structural receptor binding interactions. Much of the in vitro signaling has been previously demonstrated by a prior publication (Altindis et al, PNAS 115:2461, 2018), and some of the current findings contradict what the group found previously in terms of IR signaling and preferential stimulation of the AKT compared to ERK pathways after binding to IGF1R. The in vivo findings in the current study show that treatment of mice expressing hGH with LCDV1-VILP through AAV delivery reduces body weight without affecting blood glucose levels indicating in vivo antagonist actions. This is in contrast to their prior findings that acute injection of the peptide into mice caused a significant reduction in blood glucose (Altindis, 2018). Moreover, another publication by some of the same authors demonstrated that LCDV1-VILP has antagonistic properties for the IGF1R through its C-peptide domain (Zhang et al. Mol. Metab. 53:101316, 2021). The major new contribution in this study is in the cryo-EM structural analyses of the peptide and IGF1R binding. The value of the studies on this peptide is in the potential for understanding how to use it to develop IGF1 antagonists for use in patients.

Major Comments

1. Figure 2, panels a,b – for completeness, it is important to include the competition binding assays using the B isoform of the hIR as well as panels c-f shown for the A isoform. There is no explanation why this Figure shows mixed results for the A and B isoforms. Is it entirely clear they will be the same? Moreover, based on Altindis, 2018 it is clear the A isoform may show stronger binding.

We appreciate the reviewer's comment, however, we don't want to over-interpret the data. In previous studies, including a recent study by Chrudinova et al. (Mol. Metab., 2020), we find that while insulin binds with the similar affinity to both insulin receptor isoforms A and B (see below Figure 2A and B, left panel), dcLCDV1-VILP is not able to bind to either insulin receptor isoform with measurable affinity. Consistent with these data, in preadipocytes overexpressing either the insulin receptor isoform A or B, insulin induced receptor autophosphorylation and post-receptor signaling was similar with both receptor isoforms, and dcLCDV1-VILP had minimal effect to activate either insulin receptor isoform (Chrudinova M. et al, 2020, Mol. Metab., Figure 2A, B, D and E). Similarly, Altindis et al have shown that the autophosphorylation of both insulin receptor isoforms could be induced by scLCDV1-VILP, but only at extremely high concentrations (Altindis E. et al. 2018, PNAS, Figure 2 D and E, right panel). Thus, in terms of insulin receptor activity, both insulin and LCDV1-VILP shown very differential activity, regardless of which isoform is studied. We have now made this point clearer in the text in the revision.

Allindis, Emrah, et al. "Viral insulin-like peptides activate human insulin and IGF-1 receptor signaling: A paradigm shift for host-microbe interactions." *Proceedings of the National Academy of Sciences* 115.10 (2018): 2461-2466.

2. As the authors point out, it is interesting that the LCDV1-VILP activated Akt but showed no stimulation of ERK phosphorylation whereas IGF-1 and insulin activated both pathways. The conclusion presented is that there may be "a unique post-receptor pathway bias in the effects of these ligands". This raises the question of whether binding of the LCDV1-VILP ligands do not result in receptor internalization and ERK signaling in endosomes.

This is an interesting point raised by the reviewer. To assess the potency of LCDV1 versus IGF1 to induce receptor internalization, we have now performed an internalization assay using murine brown preadipocytes expressing the human IGF1R and using surface biotinylation to assess internalization. We find that stimulation of cells with 10nM of IGF1 induces a decrease by 57% of IGF1R abundance at the plasma membrane with 45 min. scLDV1-VILP used at concentrations of 10nM and 1000nM was also able to reduce the level of membrane IGF1R by 40% and 47% respectively, in line with its lower potency to bind and activate IGF1R. Since scLCDV1-VILP is capable of inducing IGF1R internalization, this selective signaling seems to be independent of IGF1R endocytosis and must happen at a post-receptor level. These new data are shown below and have been added as **Fig. S1a and S1b** in the manuscript.

3. Figure 3 – It is important to do this experiment in human cells expressing endogenous levels of both the IGF1R and IR. What might be the outcome based on no overexpression or on having some hybrid receptors present? Some of these studies (on other cell types) are presented in the prior Altindis publication with conflicting results to those presented here. This needs to be addressed.

This is a good point since all the signaling experiments in this manuscript were performed in cells overexpressing only one type of receptor. Therefore, we assess the impact of the VILPs on a more physiologic model, we have studied the effects on MCF7 cells, stimulated with various concentration of either IGF-1, sc- or dcLCDV1-VILP. MCF7 is a breast cell line characterized by a co-expression of insulin and IGF1 receptor. Consistent with the Figure 2a and 2c, IGF-1 is more potent in inducing phosphorylation of the IGF1R, AKT and ERK 1 and 2 compared to the sc- and dcLCDV1-VILP (see figure below). In line with our competition binding data on IGF1R and IR, scLCDV1-VILP appears to be more potent in stimulating AKT phosphorylation compared to dcLCDV1-VILP. Consistent with the Figures 3a and 3c, only scLCDV1-VILP is able to antagonize IGF1R signaling induced by 10 nM of IGF1. These new data confirm the potency on scLCDV1-VILP to antagonize IGF1R signaling in a model of cells that express physiologic and endogenous levels of IGF1R and IR and are now included in the manuscript as Figures S2.

4. Figure 4a,b – Cell number after 48 hr treatment was used to quantify proliferation. This could also be altered by changes in survival that should be included. Alternatively, a direct measure of proliferation is needed.

The reviewer raises an interesting point. To assess a potential change of survival induced by the scLCDV1-VILP, in a new experiment, we incubated murine preadipocytes overexpressing hIGF1R with a high dose of either scLCDV1-VILP or dcLCDV1-VILP. As shown in the figure on the right, neither sc- nor dc-LCDV1-VILP affect the survival of these cells. Thus, we conclude that the lower cell number seen when the cells are stimulated with a mixture of 1000 nM scLCDV1-VILP and 10 nM of IGF1 is due to a decrease of cell proliferation and not to an increase of cell death. This figure has been added to the manuscript as new Figure 4c.

5. In vivo studies on the hGH transgenic mice:

a) There is no discussion of why only female mice were used for the in vivo studies on the hGH transgenic mice. Was the effect on male mice similar?

bGH mice were bred for this study with the intent of using both male and female mice. Unfortunately, the litters born were uncharacteristically mainly female. Thus, we used what was available from this breeding. While it would be interesting to see if males had a different response, to reconstitute a new colony of mice for this experiment would require more than 6 months and involve extensive animal costs. Without a specific hypothesis, therefore, we feel it is not worth looking for sexual dimorphism at this time.

b) In these experiments, the authors include an AAV only control. Another control should be included for growth measurements to show that AAV-IGF1 itself does not have the same effect on body growth as the AAV-LCDV1.

While this is an interesting alternative, the bGH transgenic mice already over secrete IGF-1. The AAV-IGF1 idea was something we considered as an alternative, but felt it would be producing much higher levels of IGF-1, so more difficult to inhibit, and we were concerned about mice over-expressing hormones from two different AAV vectors, and competition of these for the cellular hosts.

6. A major conclusion from the structural analyses is that the LCDV1-VILP ligand binding to the IGF1R causes an inactive conformational state such that it cannot activate downstream signaling. It is then unclear how Akt is activated as shown in Figure 2.

As now discussed in the revised manuscript, we hypothesize that at low concentrations only one scLCDV1-VILP molecule may bind to the IGF1R leading to a different conformational state resulting in partial activation of the IGF1R (line 361-367).

Minor Comments

Figure 5

a) Panels a,b – the models presented here are based on prior publications. References are

provided but it is not clear if the images have been reconstructed or reproduced here with permission.

The Figures were reconstructed from PDBs, as now indicated in figure legend.

b) Panels g,h – no dashed line is apparent in the figure as indicated by the legend.

We thank the reviewer for noting this, and the figures are now updated.

Reviewer #2 (Remarks to the Author):

The manuscript by Moreau et al. characterizes single- and double-chain viral insulin-like peptides (VILPs) encoded by Lymphocystis disease virus-1 (LCDV-1). The authors show through a series of biochemical and in vivo mouse experiments that the single-chain version is a potent antagonist of insulin-like growth factor-1 receptor (IGF1R), with no or minimal effect on signaling through the insulin receptor (IR).

They then characterize the binding of scLCDV1-VILP to the ectodomain of IGF1R using cryo-electron microscopy. In contrast to the 1:1 complex observed previously between IGF1 and IGF1R, the authors find two molecules of scLCDV1-VILP bound to IGF1R in the two 1a binding sites. The peptides do not engage the two 1b sites, which has the effect of increasing the distance between the ends of FnIII-3 to 90 Å from 67 Å for the apo IGF1R. This will keep the cytoplasmic kinase domains from undergoing trans-phosphorylation, explaining how scLCDV1-VILP acts as an IGF1R antagonist.

This study presents findings that will be of great interest to those in the insulin/IGF1 signaling field.

Specific point:

One issue related to the 2:1 stoichiometry of VILP peptide to IGF1R is the 30-fold molar excess of peptide used in the cryo-EM study. How do we know that the 2:1 stoichiometry, rather than 1:1, is the relevant state in vivo?

The cryoEM study was performed with saturating concentrations of scLCDV-1 to ensure sample homogeneity. In the case of IGF-1 or IGF-2 binding to the IGF1R (including IGF1Rzip as used in this study), multiple studies have shown only one hormone molecule binding the receptor at similarly high concentrations of ligand due to strict negative co-operativity. Thus, we argue in the manuscript the 2:1 stoichiometry and overall receptor conformation is a key point of difference between scLCDV1-VILP and IGF-1 and -2 and provides the simplest explanation for the observed antagonism. As discussed in the response to Reviewer one, it is possible that low levels of occupancy, other forms may exist and result in a mix of signaling outcomes. This point has now been clarified in the manuscript discussion (see lines 372-379).

Reviewer #3 (Remarks to the Author):

Moreau and Kirk et al. have identified a viral insulin-like peptide that can act as an antagonist of IGF-1R with no/limited effects on the highly homologous insulin receptor. The cryo-EM structure reveals a novel 2:1 ligand:IGF-1R complex that explains the broad mechanism by which scLCDV1-VILP acts as an antagonist. In-vivo- experiments in mice show that expression of LCDV1-VILP causes a reversal of an IGF-1R overstimulation phenotype without adverse effects on resting blood glucose, reaffirming the therapeutic utility of an IGF-1R antagonist. This work advances our basic understanding of how IGF-1R is activated and offers valuable insight towards the development of potential IGF-1R antagonists that have high selectivity for IGF-1R vs IR.

Suggested revisions:

1. Only one of the citations on line 65 has results that show the membrane proximal spacing. Ref 20 is only the first three domains of IGF-1R.

The surplus reference has been removed.

2. On line 119, the sequence of scLCDV1-VILP and dcLCDV1-VILP are the same and therefore the dc/sc prefix should be removed

This text has been updated as suggested.

3. On line 66, the 67 A spacing applies only to IGF-1R. The spacing for IR is closer to 115 A. The wording could be changed from “these receptors” to “the IGF-1 receptor”.

This has been updated in the manuscript (line 67).

4. Figure 3B and line 179: While the effect is not statistically significant because of the large range in measured values, the average fold-change for IR-induced ERK phosphorylation (3d, right) is of similar magnitude to the ERK suppression seen in the IGF-1R case (3c right). I suggest the authors collect additional data to improve statistical power or soften the language here. Another way the authors could gain supporting data for this idea is to perform the cell growth assay in cells over-expressing IR in addition to the data already shown for cells overexpressing IGF-1R.

As mentioned by Reviewer #3, the range of values for the quantification of ERK phosphorylation is large. In order to reduce this range, we have now collected additional data with an n=6, instead of 4 previously. The new quantification for the phosphorylation is shown in Figure 3d. In addition, the text has been updated to mention the lack of significance of this change. Moreover, to confirm the absence of functional consequence of sc- and dcLCDV1-VILP interaction with the insulin receptor, we performed a proliferation (MTT assay) on the murine brown preadipocytes that express only the human insulin receptor. As shown below, neither sc nor dc-LCDV1-VILP affect the proliferation induced by 10 nM of insulin.

5. Please add units for the y-axis in Figure 3.

These figures have been updated.

6. On line 186, the phrase “the curve was shifted two orders of magnitude to the right” does not seem consistent with Fig. 4a. (1 nM to 10 nM is one order of magnitude)

This mistake has been corrected in the manuscript.

7. For Fig 4c, the magnitude of the weight loss effect is not apparent without reading about these mice. Would it make sense to show the weight of the control mice over the same period?

The average weight measured at week 2 and 6 after the AAV injection is shown, for both group side by side, in Figure 4D.

8. Line 281: the wording “binds as a monomer” is confusing as both IGF-I and scLCDV1-VILP bind to IGF-1R as monomers.

This sentence has been updated to remove ambiguity.

9. On line 366, mention of the Gly to Ser mutation in the C-peptide is the first mention within the paper. Should it say B chain instead of C-peptide?

This sentence has been updated to remove ambiguity.

10. I am glad the authors mentioned the C-peptide context being somewhat different at site1a and site 1b. This could explain the importance of the VILP C-peptide sequence for antagonism. If there are any interactions with the receptor that are expected to be different in the site 1a and site 1b states, I suggest making a supplementary figure illustrating this; however, it is not clear to me that there are such interactions apparent from the cryo-EM data without seeing the atomic coordinates, so I leave this to the discretion of the authors.

This has been addressed in response to the comment 3 and 4 of reviewer #4. Briefly, our hypotheses were based on the relative position of residues between scLCDV1-VILP and IGF-1 and not based solely on direct observations in the cryoEM map itself (beyond confirming the location and backbone structure of scLCDV1-VILP). As describing and illustrating the hypothesized key interactions also gave reviewer #4 the impression that we were claiming to have confirmed these interactions directly using the cryoEM data, we have chosen to soften our language around specific side-chain interactions and remove or adjust the corresponding figures from the manuscript.

Reviewer #4 (Remarks to the Author):

Moreau, Kirk, et al. characterize two viral insulin/IGF-1 like peptides (VILPs) using various in vitro and in vivo methods.

Using a competition binding assay they show that the single chain peptide (scLCDV1-VILP) is more potent in competing with Insulin and IGF-1 for binding their respective receptor compared to the double chain peptide (dcLCDV1-VILP). However, all determined VILP binding affinities are lower than the ones of insulin and IGF-1 to their respective receptor. scLCDV1-VILP showed the highest affinity to hIGF1R still being an order of magnitude lower than that of IGF-1 itself.

In line with these findings only scLCDV1-VILP showed a notable effect on IGF1R signaling in cell culture experiments. The authors went on to show that scLCDV1-VILP has an antagonistic effect on IGF-1 induced IGF1R signaling while it had no effect on insulin induced IR signaling. Consequently, they show that scLCDV1-VILP does inhibit IGF-1 induced cell proliferation and mouse growth.

In order to explain this antagonistic effect on a molecular level, the authors turned to cryo-EM to determine the structure of scLCDV1-VILP in complex with the ectodomain of IGF1R. Using single particle analysis, they solved the structure at a nominal resolution of 4.6 angstrom and derived an atomic model thereof. The presented structure, however, is not convincing to back up the drawn conclusions and does not live up to community standards. Therefore, I unfortunately cannot recommend publication of the manuscript in the journal Nature Communications.

Major issues

1) line 208ff and Figure 5: the novel findings from this study should be emphasized more, i.e. it should be made more clear that the model shown in panel c is the new finding. The reconstructed cryo-EM density also has to be presented in the main text and figures and not only the last supplementary figure S4 with only one view. The chosen surface representation of the derived model is misleading in regards to the claimed resolution of the cryo-EM reconstruction. In the main text, previously published structures are discussed at length before the new findings from this study are presented relatively brief. It should be the other way round.

As suggested by the reviewer, new figures and clarity around resolution heterogeneity has been added. We have also re-ordered the figures as suggested.

2) Figure S4: The relatively flat fall-off of the FSC starting from low frequencies points at issues in the particle alignment and 3D reconstruction. With such 'pathogenic' FSC curves the nominal resolution as given by the interception at 0.143 has to be taken with care and the resolution claim has to be backed up by the presence of characteristic features. At 4.6 Å, alpha-helices including the pitch have to be clearly visible. Another characteristic feature in this resolution range is the beginning separation of beta-strands in the density. Neither of those features can be observed in the shown density map, and if present, must be presented in a readily accessible way.

In order to improve the 3D reconstruction, the authors could employ 3D classification in Relion as Ab-initio reconstruction as implemented in cryoSPARC is not always suited well for classifying particles.

As suggested by the reviewer, new figures (Fig. 5a, b and fig. S4e) have been added to illustrate secondary structure observable in the cryoEM density, and clarity around resolution heterogeneity of the reconstruction added. We add that exhaustive attempts were previously made at 3D classification in addition to further focused refinements in a variety of programs and did not result in improved homogeneity or resolution of the resulting reconstruction compared to that described in the manuscript. These attempts have been acknowledged in the updated methods section.

3) line 604ff: Even though atomic coordinate model building is technically possible at 4.6 Å resolution, it is a very challenging task and requires a high-quality reconstruction. With the

above-mentioned limitations regarding map quality, I don't believe atomic coordinate model building is appropriate in this case. This is also evident from the poor atom inclusion shown on page 6 and 7 of the wwPDB EM Validation Report. The authors should limit the map interpretation to rigid-body fitting of known parts of the structure. In case atomic coordinate model building is pursued, at least a map-to-model FSC should be presented. A cut-off value is reported in Table 2, but it is not evident where this value comes from. For map-to-model FSCs, the reported cutoff should also be at 0.5 instead of 0.143.

As suggested by the reviewer, model building has been repeated, limited to rigid-body fitting of relevant domains from PDB entries 5U8R and 6PYH, with strictly restrained regularization of only the inter-domain sections to ensure backbone continuity and modest fitting of the scLCDV1-VILP backbone from an AF2 model into clear density with Ramachandran and dihedral restraints always applied. Adjustments to statistics have been included. We note atom inclusion values per chain are far better than the overall impression as domain FnIII-2' that lies mostly outside of density after NU-refinement at the chosen threshold has been left in for clarity. One figure with a complete ectodomain backbone model based on a homogeneous intermediate map (low resolution) has been left in for illustrative purposes, but clearly demarcated as such and our submitted cryoEM density included in the same figure. This has been acknowledged and updated in the manuscript.

4) Figure S2: the detailed structural analysis on the side-chain level, especially in panel d and e should be avoided as it gives the reader a false sense of model quality. In the presented cryo-EM density basically no side chain densities are visible.

Figures have been adjusted or removed and language softened as recommended by the reviewer. We acknowledge the manuscript was ambiguous as to where conclusions were drawn from (i.e. predicted peptide structures vs directly observed in cryoEM structure) and an impression of over-interpretation was reasonable. Analysis drawn from the cryoEM map has now been strictly limited to the relative domain positions of the receptor and to confirm the binding location and backbone structure of scLCDV1-VILP when combined with the AlphaFold2 predicted structure of scLCDV1-VILP. The lack of direct evidence of the hypothesized interactions in the cryoEM map has been clarified. Some speculation of interactions based on the proximity of residues has been retained but has been softened to ensure our conclusions do not give the impression of certainty.

Minor issues

*Figure 1c: it should be pointed out in the figure or the legend that the presented models for the VILPs represent AlphaFold2 predictions. A representation of the AlphaFold2 confident score (pLDDT) should also be included. PDB codes of the shown insulin and IGF-1 models should be indicated.

This has been updated in the manuscript as suggested.

*Figure 4d: the Y-axis numbering seems to be partially covered by the label.

This has been updated in the manuscript as suggested.

*Figure 5a/b: PDB codes of the already published structures should be indicated

This has been updated in the manuscript as suggested.

*Figure S3a: scale bar is missing from the micrograph

This has been updated in the manuscript as suggested.

*line 233: further refinement focusing on [...]: it is not clear from the methods section what is meant by that statement. Which density is that statement referring to? If it's not shown in the supplement yet, it should be included.

This has been updated in the manuscript as suggested.

*line 608: what is meant by 'rigid body real-space refinement'?

This has been updated in the manuscript as suggested.

Reviewer #1 (Remarks to the Author):

The authors have addressed my concerns.

Reviewer #2 (Remarks to the Author):

The authors have satisfied my concerns.

Reviewer #3 (Remarks to the Author):

All my comments were well addressed by the authors. I have no more comments to add.

Reviewer #4 (Remarks to the Author):

In the revised version of the manuscript entitled "Unique Interaction of a Viral Insulin-Like Peptide with the IGF-1 Receptor Produces A Natural Antagonist" by Moreau, Kirk et al the authors have now significantly softened conclusions based on the included cryo-EM work. They now also limit the coordinate model refinement steps to rigid-body fitting of known structures and AlphaFold2 predictions with small manual adjustments more appropriate for the presented 3D density map. Taken together, the cryo-EM part of the manuscript and conclusions based on it are now acceptable after following revisions:

1) line 238ff: delete "due to NU-refinement" from the sentence. NU-refinement is not the cause for the resolution variation. Such significant variations in local resolution/map quality can rather be attributed to flexibility, conformational and/or compositional heterogeneity inherent to the sample itself. In fact, methods like NU-refinement aim to alleviate those effects in silico.

2) Figure 5 a,b: please indicate the contour level of the displayed densities used to generate each panel. For panel b, please add a statement in the methods section on how the respective densities have been generated from the full map. E.g. was density carved out from the full map? If so, which radius around the respective atoms from the coordinate model was used?

3) add the map vs model FSC curve used to determine the 0.5 cutoff value specified in Table 2 to the supplement, e.g. in Figure S6 or as a separate Figure. A reference to it should be added accordingly to the Model building part in the Methods section.

4) add a reference to Figure S4e where appropriate, e.g. in line 242.

Reviewer #1 (Remarks to the Author):

The authors have addressed my concerns.

Reviewer #2 (Remarks to the Author):

The authors have satisfied my concerns.

Reviewer #3 (Remarks to the Author):

All my comments were well addressed by the authors. I have no more comments to add.

Reviewer #4 (Remarks to the Author):

In the revised version of the manuscript entitled "Unique Interaction of a Viral Insulin-Like Peptide with the IGF-1 Receptor Produces A Natural Antagonist" by Moreau, Kirk et al the authors have now significantly softened conclusions based on the included cryo-EM work. They now also limit the coordinate model refinement steps to rigid-body fitting of known structures and AlphaFold2 predictions with small manual adjustments more appropriate for the presented 3D density map. Taken together, the cryo-EM part of the manuscript and conclusions based on it are now acceptable after following revisions:

1) line 238ff: delete "due to NU-refinement" from the sentence. NU-refinement is not the cause for the resolution variation. Such significant variations in local resolution/map quality can rather be attributed to flexibility, conformational and/or compositional heterogeneity inherent to the sample itself. In fact, methods like NU-refinement aim to alleviate those effects in silico.

The reference to NU-refinement as a source of resolution variation was removed.

2) Figure 5 a,b: please indicate the contour level of the displayed densities used to generate each panel. For panel b, please add a statement in the methods section on how the respective densities have been generated from the full map. E.g. was density carved out from the full map? If so, which radius around the respective atoms from the coordinate model was used?

Indication of map densities generation was added in the methods section (line 658-660).

3) add the map vs model FSC curve used to determine the 0.5 cutoff value specified in Table 2 to the supplement, e.g. in Figure S6 or as a separate Figure. A reference to it should be added accordingly to the Model building part in the Methods section.

The map vs model FSC curve was added in the manuscript as supplementary Fig. 4f.

4) add a reference to Figure S4e where appropriate, e.g. in line 242.

The reference to the supplementary Fig 4e was added in line 241.

We thanks all the reviewers for their positive feedback.